

# Constraining Chemical Transport PM$_{2.5}$ Modeling Using Surface Monitor Measurements and Satellite Retrievals: Application over the San Joaquin Valley

Mariel D. Friberg[1, 2], Ralph A. Kahn[1], James A. Limbacher[1, 3], K. Wyat Appel[4], James A. Mulholland[2]

[1]NASA Goddard Space Flight Center, Greenbelt, MD 20771, USA
[2]School of Civil & Environmental Engineering, Georgia Institute of Technology, Atlanta, GA 30332, USA
[3]Science Systems and Applications Inc., Lanham, MD 20706, USA
[4]US EPA, Research Triangle Park, NC 27711, USA

*Correspondence to*: Mariel D. Friberg (mariel.d.friberg@nasa.gov)

**Abstract.** Advances in satellite retrieval of aerosol type can improve the accuracy of near-surface air quality characterization, by providing broad regional context. In addition to aerosol optical depth, qualitative constraints on aerosol size, shape, and single-scattering albedo provided by multi-angle instruments, such as the Multi-angle Imaging SpectroRadiometer (MISR) on the NASA Earth Observing System's Terra satellite, can provide frequent, spatially extensive, instantaneous constraints on chemical transport models (CTMs), which can be especially useful in areas away

from ground monitors and progressively downwind of emission sources. CTMs (e.g. the Community Multi-scale Air Quality Modeling System) complement such data by providing complete spatial and temporal coverage, offering additional physical constraints (e.g., conservation of aerosol mass, meteorological consistency) independent of observations, and aid in identifying relationships between observed species concentrations and emission sources. Incorporating satellite aerosol information in the development of PM$_{2.5}$ concentration metrics can lead to a decrease in metric uncertainties and errors.

This work focuses on the degree to which regional-scale satellite and CTM data can be combined to improve surface estimates of PM$_{2.5}$, its major chemical component species estimates, and related estimates of uncertainty. Aerosol airmass types over populated regions of Southern California are characterized using satellite data acquired during the 2013 San Joaquin field deployment of the NASA DISCOVER-AQ project. Using the MISR Research Aerosol retrieval algorithm

(RA), we investigate and evaluate the optimal application of incorporating 275 m horizontal-resolution aerosol airmass-type maps and total-column aerosol optical depth into a 2 km resolution, regional-scale CTM output, to obtain constrained fields of surface PM$_{2.5}$. Contemporaneous surface observations are used to evaluate the results. The impact of incorporating MISR aerosol data on the ability to characterize air quality progressively downwind of large single sources is discussed. The spatiotemporal R$^2$ values for the model, constrained by both satellite and surface-monitor measurements based on 10%

withholding, are 0.79 for PM$_{2.5}$, 0.88 for NO$_3^-$, 0.78 for SO$_4^{2-}$, 1.00 for NH$_4^+$, 0.73 for OC, and 0.31 for EC. Regional cross-validation temporal and spatiotemporal R$^2$ results for the satellite-based PM$_{2.5}$ improve by 30% and 13%, respectively, in





comparison to CTM simulations, and provide finer spatial resolution. $SO_4^{2-}$ cross-validation values showed the largest spatial and spatiotemporal $R^2$ improvement with a 43% increase. Assessing this technique in a well-instrumented region opens the possibility of using the satellite data to apply the technique globally.

## 1 Introduction

To investigate air pollution health effects on humans, population-based epidemiologic time-series studies often use exposure measures derived from regulatory monitoring networks (Laden et al., 2006; Pope et al., 2009; Özkaynak et al., 2009). Even for the continental US, many ambient, ground-level fine particulate matter ($PM_{2.5}$) chemical datasets are acquired only once every three or six days, and at many sites for less than a decade or two. In addition, the monitors tend to be concentrated in a small number of populated counties, with the exception of the Interagency Monitoring of Projected Visual Environment

(IMPROVE) program sites located primarily in US national parks (Hand et al., 2011). Prior to 2009, instrument types and sensitivities varied considerably from monitor to monitor and among monitoring networks (Chow et al., 2010), making comparisons and uncertainty assessments difficult.

   Urban-level epidemiological time-series studies often span large geographic regions (Goldstein and Landovitz, 1977; Wade

et al., 2006). Especially for long-term exposure analysis, broad regions within or downwind of urban and industrial centers are also of concern due to the presence of distributed populations, and natural and agricultural ecosystems. Reducing exposure-metric error caused by inadequately characterized spatial variability, which is often much larger than instrument error, can substantially reduce bias and improve precision in epidemiologic results (Ito and Thurston, 1995; Pinto et al., 2004; Goldman et al., 2012). This is particularly relevant for regional-scale studies, where the spatial variability (e.g. urban-

to-rural gradients) of ambient surface $PM_{2.5}$ and chemical species concentrations, which is fundamental to effectively conducting environmental epidemiologic studies and air quality assessments, can be lacking.

   Although chemical transport model (CTM) simulations provide more complete spatial and temporal coverage than surface monitors, they rely on uncertain inputs about pollution source characteristics that can contain significant biases. The

accuracy of the simulated fields is also affected by the accuracy of the simulated meteorology, and of the physical and chemical parameters specified in the model (Tong and Mauzerall, 2006). Errors in these fields can be identified and sometimes quantified by comparison with coincident ground- and aircraft-based observations. Under satisfactory retrieval conditions, satellite-derived aerosol optical depth (AOD), atmospheric scattering, light absorption, and extinction by suspended particles can be leveraged to constrain the columnar CTM simulations in sparsely monitored areas.





Early PM$_{2.5}$ air quality studies that were based upon space-based data simply correlated ground-level PM$_{2.5}$ concentrations and satellite-derived AOD from the MODerate resolution Imaging Spectroradiometer (MODIS) instruments, without accounting for particle vertical distribution, overlooking the day-to-day variations, and/or neglecting any consideration of aerosol speciation (Chu et al., 2003; Wang and Christopher, 2003; Engel-Cox et al., 2004; Chu, 2006; Gupta and

Christopher, 2009; Wallace and Kanaroglou, 2007; Schaap et al., 2009; Zhang et al., 2009; Hu and Rao, 2009; Tsai et al., 2011; Hu et al., 2014). This approach worked well for PM$_{2.5}$ when the aerosol was almost entirely concentrated in the near-surface boundary layer, but suffered when transported aerosol made a significant contribution to the total-column AOD, or when the boundary layer was deep or variable on short timescales. Other early studies used surface measurements (Al-Saadi et al., 2005) or CTMs (Liu et al., 2004; Koelemeijer et al., 2006; Mathur, 2008; Van Donkelaar et al., 2010; Drury et al.,

2010; Wang et al., 2010; Donkelaar et al., 2013; Boys et al., 2014; Ma et al., 2014) to provide some constraint on aerosol vertical distribution, but still did not account in detail for either spatial or temporal variations of the relationship between total-column AOD amounts and surface PM$_{2.5}$ concentrations, and provided no observational constraints on aerosol type. The Van Donkelaar et al. (2010) study used space-based CALIPSO lidar backscatter profiles to validate the GEOS-Chem model vertical distributions globally, aggregated over a four-year period. Advanced statistical models that use land-use,

meteorological, and relative humidity parameters have been applied to increase the accuracy of AOD-to-PM$_{2.5}$ estimates (Kumar et al., 2007; Di Nicolantonio et al., 2009; Lee et al., 2011; Kloog et al., 2012; Hu et al., 2014; Ma et al., 2014; Song et al., 2014; Lv et al., 2016; Ma et al., 2016). Several of these models are location specific and assume limited spatial variability in the relationship, which affect the model as the domain becomes larger.

The first papers to include some space-based aerosol type information along with AOD from satellites for air quality applications used the Multi-angle Imaging SpectroRadiometer (MISR) spherical vs. non-spherical distinctions to separate airborne dust from spherical particles over the continental US, and constrained aerosol vertical distribution and speciated the spherical components with an aerosol transport model (Liu et al., 2007a; 2007b). Subsequent work applied MISR aerosol size and shape constraints over the Indian subcontinent and surrounding areas to map seasonal changes in aerosol type (Dey

et al., 2012) and combined MISR particle shape and qualitative light absorption information to make a first effort at mapping aerosol airmass types over an urban area, i.e., Mexico City (Patadia et al., 2013).

To estimate ambient PM$_{2.5}$ mass and associated chemically speciated concentrations on regional scales, a systematic and practical approach is developed and evaluated in this paper. The approach uses ground-based PM$_{2.5}$ measurements, where

available, to anchor speciated, near-surface CTM aerosol concentrations. To help constrain the model over extended regions, MISR total-column AOD is also applied, along with qualitative, column-effective aerosol type observations (i.e., size, shape, and single scattering albedo) where mid-visible AOD values exceed 0.15. Enhanced aerosol-type retrievals from





the MISR Research Aerosol (RA) retrieval algorithm (Kahn et al., 2001; Limbacher and Kahn, 2014), at 1 km horizontal resolution, are at the heart of this new approach.

To demonstrate the method, we apply it over a case study area in the San Joaquin Valley of California during the DISCOVER-AQ field campaign in this region, on six days when there is good MISR coverage. The results account for spatiotemporal variability in $PM_{2.5}$ and the chemical component concentrations. The accuracy of estimated concentrations and evaluation of the latest MISR-RA ability to typify urban AOD, aerosol mixtures, and aerosol airmasses, are examined by comparing the results with speciated ground observations and standard model-fitting statistics. Section 2 describes the datasets involved, Sect. 3 describes the method and technical approach, and Sect. 4 presents results and validation for our
test cases. Conclusions, along with a brief discussion of prospects for wider application of this approach, are given in Sect. 5, and detailed data and ancillary documentation are provided in Supplemental Material.

## 2 Study Domain and Datasets

### 2.1 Study Domain

The San Joaquin Valley (SJV), which comprises the southern two-thirds of California's Central Valley (about 26,000 $km^2$),
has long suffered from severe air pollution issues and is among the most studied air sheds in the US (Ngo et al., 2010; Chow et al., 2006). The SJV has complex topography and meteorology, particularly in winter, when low planetary boundary layer (PBL) heights and high pollutant mixing ratios create a challenging environment for chemical transport modeling (Appel et al., 2017). This region is surrounded by the Sierra Nevada to the east, the Diablo and Temblor Ranges to the west, the Tehachapi Mountains to the south, and the Sacramento Valley to the north (Fig. 1). Although primarily a rural area, the
eight counties that comprise the SJV are home to more than 4 million residents. Despite of the semi-arid climate, the SJV is one of the world's most productive agricultural regions (Schoups et al., 2005). The SJV air shed frequently experiences high $PM_{2.5}$ concentrations during the winter due to the combination of relatively dry climate, shallow PBL heights, local source emissions, and the surrounding mountain ranges. The SJV has been in violation of the $PM_{2.5}$ National Ambient Air Quality Standards for $PM_{2.5}$ annual standard since their inception in 1997, and is the largest $PM_{2.5}$ nonattainment area in the
continental US (EPA, 2012).

The study period for this work, January and February 2013, was selected to coincide with the Deriving Information on Surface Conditions from Column and Vertically Resolved Observations Relevant to Air Quality (DISCOVER-AQ; http://discover-aq.larc.nasa.gov) field campaign. This campaign was a joint collaboration between NASA, NOAA, US EPA,
multiple universities, and several local organizations, with the goal of characterizing air quality in urban areas using satellite, aircraft, vertical profiler and ground-based measurements. Targeting the 2013 DISCOVER-AQ period for this study





provides considerable ground- and aircraft-based measurements for aerosols and fine particulate matter, which we apply as model constraints and for evaluation.

We analyze data for six days during the DISCOVER-AQ period for which (1) MISR observations were made over the study
region, (2) coincident ground and aircraft observations were acquired, including extensive field-campaign data, and (3) the key observational requirements of relatively cloud-free conditions and the presence of aerosols from different sources are met.  Of the six days for which we have MISR coverage, the mid-visible AOD exceeds 0.15 on three days: January 20[th], February 3[rd], and February 5[th]. On lower-AOD days, MISR aerosol type information has higher uncertainty for the current application and thus the analysis of speciated $PM_{2.5}$ focuses on the higher-AOD days.  Of the three higher-AOD days,
January 20[th] has the least cloud cover, followed by February 5[th], so these days will be the main focus of detailed analysis. The method developed here can in the future be applied to many other polluted regions of the world where AOD exceeding 0.15 is common, such as South and East Asia, North Africa, and many major metropolitan areas.

The ground-based, aircraft, and simulation data used in this study are described briefly in the rest of this section, along with
the MISR-RA retrieval product.

### 2.2 Ground-based PM Mass and Speciated Measurements

This study focuses on $PM_{2.5}$ mass and the five components that dominate total $PM_{2.5}$ in the SJV: sulfate ($SO_4$), nitrate ($NO_3$), ammonium ($NH_4$), elemental carbon (EC), and organic carbon (OC).  Data files of ambient aerosol particulate matter species concentrations for sites within the SJV for January and February 2013 were obtained from two EPA sources: (1) daily
averaged $PM_{2.5}$ Federal Reference Method (FRM) and Federal Equivalence Method (FEM) mass from the Air Quality System (AQS; https://www.epa.gov/aqs), and (2) daily averaged total $PM_{2.5}$ and chemically speciated mass (measurements typically made every third or sixth day) from the Chemical Speciation Network (CSN; Solomon et al., 2014).

FRM compliant data from gravimetric filter-based samplers and FEM compliant data from continuous mass monitors
provide spatial variability of $PM_{2.5}$ mass (EPA, 2004).  The $PM_{2.5}$ FRM mass is determined gravimetrically by weighing particles on filters pre- and post-deployment.   They are equilibrated at a constant relative humidity (30-40%) and temperature (20-23°C).  Monitor locations are shown in Fig. 1, and Table 1 lists monitor summary statistics.  Daily $PM_{2.5}$ concentrations measured by the FRM method are considered $PM_{2.5}$ ground truth, i.e., their uncertainties are small compared to those of the other $PM_{2.5}$ values used in this study.





### 2.3 DISCOVER-AQ AERONET DRAGON

The AErosol RObotic NETwork (AERONET; Holben et al., 1998) has 10 permanent sun photometer (SP) monitors operating in the study region. During the DISCOVER-AQ 2013 deployment, these monitors were supplemented with an additional 14 temporary monitors termed the Distributed Regional Aerosol Gridded Observation Network (DRAGON) to

provide a more regionally dense dataset for satellite validation and *in situ* comparisons (Fig. 1). AERONET/DRAGON SPs measure AOD in multiple spectral bands from the ultraviolet (UV, ~340 nm) to the near-infrared (NIR, ~ 1640 nm), with an accuracy within ±0.015 (Eck et al., 1999).

We use version 2 (v2) level 2 (L2) AERONET/DRAGON AOD and Angstrom Exponent (ANG) data for the six study days.

The L2 data were sun-calibrated after field deployment, cloud screened (Smirnov et al., 2000), and quality controlled. The AOD at 550nm wavelength is calculated using a quadratic log-log fit to AERONET observations at shorter and longer wavelengths (Eck et al., 1999). Columnar AODs at 550nm derived from AERONET are considered as AOD ground truth in this study.

### 2.4 Chemical Transport Model Simulations

Simulations of the coupled Weather Research and Forecasting model (WRF; Skamarock et al., 2008), version 3.4, and the Community Multiscale Air Quality model (Byun and Schere, 2006), version 5.0.2, were obtained from the US Environmental Protection Agency (EPA). These atmospheric simulations, at 2 km × 2 km horizontal grid spacing with 35 vertical layers, cover the entire SJV and surrounding major cities during the months of January and February of 2013. Concentration fields from the fixed 2 km × 2 km horizontal CMAQ grid were downscaled to a horizontal grid of 275 m x

275 m by linear interpolation and used as the reference grid for all subsequent analyses. Emission data were based on the 2011 EPA National Emissions Inventory (EPA, 2015) with 2013 updates to electric generating unit emissions, fire, and mobile sources. Biogenic emissions were generated in-line to CMAQ using the Biogenic Emissions Inventory System (BEIS; http://www.cmascenter.org) version 3.14, and the emissions were processed using the Sparse Matrix Operator Kernel Emissions (SMOKE; Houyoux et al., 2000) version 3.5. The carbon bond 2005 chemical mechanism used was CB05TULC

(Yarwood et al., 2005; Whitten et al., 2010; Sarwar et al., 2012). The lateral Boundary Conditions (BCs) for the 2 km simulation were derived from a coupled WRF-CMAQ simulation with 4 km × 4 km horizontal grid spacing, covering the entire state of California and the surrounding areas. Boundary conditions for the 4 km simulation were derived from a 36 km simulation covering the contiguous United States, and BCs for the 36 km simulation were provided by a GEOS-Chem simulation (Bey et al., 2001) with the chemical species mapped to the corresponding CMAQ species (Appel et al., 2017).

The EPA conducted a model evaluation of CMAQ v5.0.2 with respect to the scientific updates to v5.1 (Appel et al., 2017). In that study, fine particulate matter simulations were biased low compared to observed concentrations over the SJV during





the winter months. Winter $PM_{2.5}$ average mean bias (Model - Observations) in the SJV exceeded -10 µg m$^{-3}$. Errors in simulated PBL height and mixing were considered to be contributing factors to the January $PM_{2.5}$ underestimation in the SJV. Although CMAQv5.0.2 is missing several secondary organic aerosol species of anthropogenic volatile organic carbon (i.e., long-chain alkanes and naphthalene) in its aerosol module (AERO6 v5.0.2), the mass contribution of these species to

$PM_{2.5}$ during the winter was minimal (less than ±0.5 µg m$^{-3}$) in the SJV (Appel et al., 2017). At the time this study was conducted, CMAQ v5.1 results were not yet available.

### 2.5 Satellite Observations

The primary satellite resource for this study is the MISR instrument. We supplement the MISR-RA aerosol data with results from the MODIS instruments. They offer more extensive spatial coverage and provide up to two observations per day (one

in the morning, and one in the early afternoon), though with larger AOD uncertainty over land, and with no constraints on aerosol type over land (Levy et al., 2013). We describe these two data sources below.

### 2.5.1 MISR – Research Aerosol Retrieval Algorithm (RA)

MISR was launched along with the first MODIS instrument aboard Terra, the flagship satellite of NASA's Earth Observing System (EOS), in December 1999 (Diner et al., 1998). Since then, Terra has maintained a sun-synchronous orbit,

descending from North-to-South over the equator at a local time of ~10:30 AM. MISR measures upwelling short-wave radiance from Earth at nine distinct view angles along the line-of-flight (±70.5°, ±60.0°, ±45.6°, ±26.1°, and nadir), in each of the four spectral bands centered at 446, 558, 672, and 866 nm. The one nadir, four forward, and four aft-viewing push-broom cameras take approximately 7 minutes to image a given 380 km wide swath of Earth. Due to swath size, it takes MISR about a week to obtain global coverage. Owing to its multi-spectral, multi-angular capabilities, high spatial resolution

(up to 275 m), and highly accurate radiometric calibration (Bruegge et al., 2007; Limbacher and Kahn, 2015; 2017), the MISR-RA is uniquely capable of supporting air-quality applications by providing information about aerosol microphysical properties at regional scales.

High-resolution (275 m) results from the MISR-RA retrieval algorithm are used to constrain aerosol concentration and type

for the CMAQ model. Because of MISR's ability to sample over a large range of scattering angles (i.e., between about 60° and 160° at midlatitudes), the RA provides column-averaged information regarding aerosol properties under favorable retrieval conditions (i.e., cloud-free, low surface albedo, mid-visible AOD exceeding about 0.15) (Kahn and Gaitley, 2015; Kahn et al., 2010). This information amounts to constraints on particle shape (non-spherical (dust) vs. spherical AOD fraction), particle size (typically three-to-five bins, e.g., "small," "medium," and "large" AOD fraction, parameterized as the

Angstrom Exponent, ANG), and particle light absorption (typically two-to-four bins, e.g., "dirty" and "clean," represented as single-scattering albedo, SSA = 1.0 – [absorbing AOD]/AOD). Although passive satellite remote sensing can only provide



information about aerosol type in two dimensions (column-averaged), a chemical transport model can be used to apportion the amount of aerosol near the surface (e.g., Liu et al., 2007a; van Donkelaar et al., 2010; this study). A brief summary of the MISR-RA retrieval process is provided in the supplemental Sect. S1.

Following the work of Patadia et al. (2013), we identify different aerosol airmasses by categorizing aerosol based on the qualitative particle size, shape, and light-absorption constraints described above. Specifically, for the purposes of this paper, the 14 aerosol components used by all 774 mixtures in the refined RA aerosol climatology (Limbacher and Kahn, 2014) can be organized into three broad aerosol-type "groups": spherical light-absorbing, spherical non-absorbing, and non-spherical (cirrus is ignored). Especially at low-AOD, the MISR-derived aerosol-type sensitivity amounts to no more than these three

groupings (Kahn and Gaitley, 2015). However, the general microphysical properties of the three broad aerosol groups (AG) can be associated with specific chemical species identified in the chemical transport model results, as described below in Sect. 3.2. From the point-of-view of retrieval sensitivity, these three categories map to common aerosol species as follows (Table S2): (1) Light-Absorbing Carbon (LAC), (2) Inorganic Ions (II) plus Organic Matter (OM) plus Sea-Salt (SS), and (3) dust. Section S2 in supplemental provides a description of how the aggregated AOD retrieval are computed for the spherical

absorbing aerosol components, and separately for spherical non-absorbing aerosol components. Over the last several years, substantial research has been published indicating that MISR AOD retrievals suffer biases in the presence of clouds (Witek et al., 2013; Shi et al., 2014; Limbacher and Kahn, 2015). Consistent with both Witek et al. (2013) and Limbacher and Kahn (2015), we present results only for days where clouds cover less than 30% of the scene within the SJV as indicated by our cloud mask, excluding the county areas that extend into the Sierra Nevada mountains.

### 2.5.2 MODIS - MAIAC

To supplement the MISR AOD values where MISR coverage is lacking, we adopt results from the MODIS Multi-Angle Implementation of Atmospheric Correction (MAIAC) advanced algorithm (Lyapustin et al., 2012), which uses time-series analysis and a combination of pixel- and image-based processing to improve the accuracy of cloud detection, aerosol

retrievals and atmospheric correction for surface retrievals. The following is a brief overview of the MAIAC Collection 6 (C6) version 2.0 (v2) June 2017 North America release aerosol product. The current study uses the MAIAC Atmospheric Properties Products (MCD19A2), which provide AOD at 0.55 µm. A more detailed description of the MAIAC theoretical background and processing steps can be found in Lyapustin et al. (2011a,b; 2012).

After extensive characterization of the MODIS-observed surface background, the MODIS Level 1B data are gridded to a fixed sinusoidal projection at 1 km horizontal resolution in order to observe the same grid cell over time. Working with a fixed grid not only facilitates the use of polar-orbiting observations as if they were "geostationary," it also simplifies




comparison of these datasets to fixed-grid model results and other measurements. In addition to the MODIS instrument on the Terra satellite, a second MODIS flies aboard NASA's Aqua satellite, which crosses the equator on the dayside at 1:30 PM local time. As a consequence of residual de-trending and MODIS C6 Terra-to-Aqua cross-calibration (Lyapustin et al., 2014), MAIAC currently processes MODIS C6 Terra and Aqua jointly as a single sensor. In addition to considerably greater

spatial coverage than MISR, this joint product offers some diurnal spread in sampling relative to the MISR snapshots.

For the time series analysis, MAIAC utilizes a 4–16 day sliding window technique of scenes from multiple MODIS overpasses to retrieve the surface Bi-directional Reflectance Distribution Function (BRDF; 0.466 μm), and spectral regression coefficients (SRCs; 0.466 μm and 2.13 μm), allowing MAIAC to retrieve AOD at 1 km spatial resolution. Unlike

instruments that collect nearly simultaneous observations using push broom scanning, the MAIAC algorithm uses the sliding window technique of consecutive clear MODIS cross-track scanned scenes (i.e., cloud-free conditions with relatively low AOD) over several days to acquire multi-angle sets of observations for each location. This allows MAIAC to retrieve the BRDF from an accumulated, multi-angle set of observations. Working under the assumption that surface reflectance changes rapidly over space and slowly over time (e.g., seasonal changes) helps the MAIAC internal dynamic land-water-snow

classification. The algorithm produces well-characterized surface reflectance that improves cloud masking, and outperforms traditional pixel-level cloud detection techniques that rely on spatiotemporal analysis (Kloog et al., 2014).

Although AOD is originally retrieved in the MODIS Blue band B3 at 0.47μm, MAIAC offers a standardized and validated AOD product at 0.55μm. With the exception of smoke and dust aerosol detection, the current algorithm does not retrieve

AOD over surfaces occurring at altitudes higher than 3.5 km. Like many satellite-based aerosol retrievals, MAIAC retrievals are unreliable for very low AOD conditions, over mountainous terrain, and over surfaces with very high albedo. The retrieval conditions that affect this study include low AOD and some cloud-contaminated scenes.

### 3 Methods

Air quality ground observations are spatially sparse, and are often temporally incomplete. CTM simulations provide

information that is independent of these observations, and are consistent with meteorology and assumed emissions. But they can contain biases, and can have difficulty capturing the spatial structure of aerosol dispersion downwind of sources. Satellites offer spatially extensive, mainly column-effective aerosol amount and type, that, if included appropriately, can reduce or eliminate fused-model-surface-measurement biases over large areas, especially regions far from concentrated surface monitors. As there are gaps in the satellite products due to clouds and other retrieval-related issues, we use the

model to help complete variable fields at several stages of the process. We also use the model to estimate the near-surface components of column-effective satellite values, and use ground-monitor data to constrain and to evaluate the results.



Our approach to fusing surface and satellite-based observations with CMAQ simulations involves five steps, illustrated in Fig. 2. First, total-column AOD and groupings of model aerosol species that match the spherical light absorbing, spherical non-absorbing, and non-spherical satellite aerosol-type AG are reconstructed from the simulated datasets. Note that the left

side of Fig. 2 tracks the process for deriving total $PM_{2.5}$, whereas the right side presents the flow for speciated $PM_{2.5}$. Blue and orange asterisks in Fig. 2 indicate where uncertainties are estimated by comparison with AERONET and the EPA ground monitors, respectively. In Step 2, spatially complete AOD and grouped AOD maps are produced for each of the six study days by combining MISR and MAIAC satellite retrievals with scaled values of the modeled AOD and AG AOD products from Step 1, respectively, to fill any remaining gaps. In Step 3, we reconstruct $PM_{2.5}$ Mass FRM from the

simulated concentration dataset. Step 4 deconstructs the satellite-based total-column AOD and grouped AOD to surface $PM_{2.5}$ and grouped $PM_{2.5}$ mass concentrations using the CTM speciated vertical distributions, respectively. The fifth and final step involves blending daily averaged ambient ground observations and satellite-based total and grouped $PM_{2.5}$ mass concentrations to estimate daily, spatially refined $PM_{2.5}$ mass and speciated pollutant concentrations.

Overall, the inputs are the speciated ground-monitor data, satellite AOD snapshots and AOD grouped by aerosol type, and the CMAQ model simulations. The outputs are the fused ground-monitor, satellite, plus model $PM_{2.5}$ mass concentration field, and speciated versions of this field. A detailed description of the key steps follows.

### 3.1 Step 1 – CMAQ- and Surface-derived $PM_{2.5}$ Using Reconstruction Method

A commonly applied $PM_{2.5}$ mass reconstruction (RM) method, also termed mass closure or material balance, is used to
compare the sum of major aerosol components to gravimetrically measured $PM_{2.5}$. This approach also accounts for unmeasured or non-simulated species to avoid double counting. Beginning with Countess et al. (1980), the RM method is used to evaluate measurements, characterize spatiotemporal chemical gradients, estimate source contributions to PM, and calculate visibility impairment due to near-surface aerosol. Additionally, the reconstructed $PM_{2.5}$ mass provides insight into the spatial variations among the speciated data (Frank, 2006; Hand et al., 2011; 2014; Malm et al., 2011). The development
of this method, along with the differences between reconstructed and gravimetric mass in the CSN and IMPROVE data sets, have been extensively studied in the US (Malm et al., 2011). Chow et al. (2015) provides a detailed literature review of the various mass reconstruction equations.

For the purposes of this study, the RM equation focuses on the following five representative chemical components, with the
relevant references cited: (1) inorganic ions (Chow et al., 1994; Chow and Egami, 1997; Andrews et al., 2000; Nolte et al., 2015); (2) organic matter (DeBell et al., 2006; Hand et al., 2011); (3) Elemental Carbon (EC), also referred to as light absorbing carbon (Bond and Bergstrom, 2006); (4) crustal material, which includes mineral and soil particles, referred to





herein as dust (Malm et al., 1994; 2011); (5) sea salt (Hand et al., 2011); and (6) other elements (Simon et al., 2011), which, in the SJV during the study period, made a negligible contribution to $PM_{2.5}$. The respective references provide details as to how multipliers for each species were derived, and summarize the evaluation performed for each major PM component.

In addition to the measured aerosol species of interest, WRF-CMAQ model outputs for relative humidity, temperature, and speciated aerosol vertical distribution were used in the $PM_{2.5}$ mass reconstruction and as needed in the other analysis steps described hereafter. The RM method, excluding negligible "other" elements, was used to compare ground observations, CMAQ results, and satellite-derived concentrations. Table S1 in supplemental material provides a summary of the aerosol equations used for the ground monitor data and CMAQv5.0.2 simulations. The RM equation used is as follows (Eq. (A) in
Chow et al., 2015):

$$RM [\mu g \ m^{-3}] =$$
$$\underbrace{[SO_4^-] + [NH_4^+] + [NO_3^=]}_{Inorganic \ Ions} + \underbrace{1.8[OC]}_{\substack{Organic \\ Matter}} + \underbrace{[EC]}_{\substack{Light \\ Absorbing \\ Carbon}} + \underbrace{1.8[Cl^-]}_{\substack{Sea \\ Salt}} + \underbrace{2.2[Al] + 2.49[Si] + 1.63[Ca] + 1.94[Ti] + 2.42[Fe]}_{Dust}$$

(1)

For each of the major chemical components involved, Chow et al. 2015 covers in detail the factors and assumptions required for the RM calculation, and those contributing to the comparison with gravimetric mass measurements. These factors include the OM/OC ratio assumptions, carbon sampling and analysis artifacts, ammonium and nitrate volatilization, limitations of using chloride to estimate sea salt content, and water retention by hygroscopic species on filters (Andrews et al., 2000; Rees
et al., 2004; Tanner et al., 2004; El-Zanan et al., 2005). Using Eq. (1) to estimate OM from OC for CMAQ output allows for consistency with satellite-derived estimates; in the future we may expand the method to include various organic aerosol species explicitly in case where we have more *in situ* data.

Following the framework of Eq. (1), the reconstructed $PM_{2.5}$ mass does not account for the positive and negative factors that
affect gravimetric and speciated measurements (DeBell et al., 2006; Frank, 2006; Hand et al., 2011; Chow et al., 2015). To close the mass-balance difference between $PM_{2.5}$ FRM gravimetric mass and ambient mass (simulated and measured), the material balance Eq. (1) was adjusted to account for factors affecting gravimetric measurements (Eq. (10) in Frank, et al., 2006).

$$PM_{2.5FRM} \ [\mu g \ m^{-3}] = RM - ([NH_4^+]_{loss} + [NO_3^=]_{loss}) + [PBW] + [Blank_{FRM}] \qquad (2)$$

where ammonium and nitrate volatilization are not captured by gravimetric measurements and thus, are accounted as negative artifacts. The particle bound water (PBW) is the water retained on the filter when particles are sampled and weighed for mass concentration. This concentration is dependent on ionic composition and relative humidity dependent species





equilibrium prior to laboratory weighing. *Blank*$_{FRM}$ accounts for the passively collected mass value on "blank" filters. The limitations and uncertainties of the reconstruction method broken down by major chemical components are discussed in detail elsewhere (Frank, 2006; Chow et al., 2015). The uncertainty estimated for the CMAQ- and satellite-based surface concentrations are discussed in Sect. 4.

**3.2 Step 2 – CMAQ-based Columnar AOD and AOD Subcategorized into Species-related Groups Derived Using the Reconstructed Extinction Coefficient Method**

Section 3.1 summarized the method applied to calculate the five representative component surface mass concentrations from the surface observations; these components are also used to derive total-column AOD from CMAQ ($\tau_{CMAQ}$). First proposed by Malm et al. (1994), the reconstructed extinction coefficient method was designed to investigate the spatial and temporal

variability of haze and visibility impairment in the US as part of IMPROVE. Since then this method has been continuously upgraded by several researchers (Malm et al., 1994; 2000; 2011; Song et al., 2008; Park et al., 2011). The process estimates extinction AOD using simulated concentrations of II, OM, SS, LAC, and Dust (Table S1) assuming externally mixed aerosols with respect to the modeled altitude (z), as follows:

$$\tau = \int \left\{ \underbrace{\sum_i \omega_i \beta_{de,i} f_{RH(z),i} C_{(z)i}}_{particle\ scattering\ efficiency} + \underbrace{\sum_i (1 - \omega_i) \beta_{de,i} f_{RH(z),i} C_{(z)i}}_{particle\ absorption\ efficiency} \right\} dz \qquad (3)$$

where

$\tau$ = aerosol extinction optical depth (AOD) at 550 nm

$i$ = chemical component

$\omega$ = single scattering albedo (SSA)

$\beta_{de}$ = specific dry extinction efficiency per mass [m$^2$ g$^{-1}$]

$f_{rh}$ = hygroscopic growth factors as a function of height

$C$ = concentration of chemical component $i$ as a function of height [g m$^{-3}$]

Equation 3 is further subdivided for dust by size in accordance with the CMAQ Aitken, accumulation, and coarse particles size categories (Park et al., 2011). The empirically based factors and their respective literature sources are summarized in

Table S3. The WRF simulated relative humidity data, *rh(z)*, were used to evaluate the height-dependent hygroscopic growth factors. The ambient particle extinction per unit length is the sum of the ambient scattering and absorption per unit length, which are the two terms in Eq. (3). When integrated over a horizontal path, the extinction per unit length is sometimes called the visibility, typically reported in Mm$^{-1}$. From Eq. 3, the dimensionless extinction AOD is obtained by multiplying the ambient particle extinction per unit length by the vertical atmospheric path length of each CMAQ layer. These are added

vertically to obtain columnar AOD values, which are compared to ground- and satellite-based AOD values in the following subsections to assess uncertainties.



The three CMAQ-based AOD AG (i.e., LAC, II+OM+SS, and Dust), indicated in Table S2, are calculated using the five major chemical components derived in Eq. (1). The CMAQ-based total-column AOD AG aggregate is equivalent to the total-column CMAQ-based AOD. Assessment of the uncertainties in these quantities, using a combination of ground-based

and satellite total-column measurements, is given in Sect. 3.4 below.

### 3.3 Step 3 – Gap-Filled Satellite-derived AOD and Grouped AOD, Using Scaled CMAQ-based AOD

To obtain a spatially complete AOD map for each case-study day, we combine the MISR-retrieved, MAIAC-retrieved, and CMAQ-based reconstructed AOD products, as CMAQ can simulate values in all grid boxes, regardless of cloud cover, surface brightness, terrain, and aerosol optical thickness. The most relevant factor affecting spatially complete satellite-

10 retrieved AOD in this study is missing retrievals due to the presence of clouds. The combined AOD product, referred herein as $\tau_{FillSAT}$, is more complete than the MISR or MAIAC products alone.

A unique component of this work involves the use of the MISR-RA aerosol species-specific groups. The MISR-RA and MODIS-based MAIAC satellite-retrieved products were combined to improve spatial coverage. Before combining retrieved

AOD products, the MISR-RA AOD at 558nm was converted to 550 nm using the retrieved ANG product. The MAIAC maps were downscaled and spatially interpolated (via bilinear interpolation) to match the downscaled model 275 m × 275 m grid. Because CMAQ and MAIAC have fixed sampling grids at each location, whereas the MISR grid varies when the same location of observed from different sub-spacecraft paths, the 275 m × 275 m MISR maps were also re-gridded to match the downscaled model grid. On days when multiple Aqua and Terra MAIAC C6v2 1 km AOD retrievals were available, the

MAIAC-Aqua AOD retrievals were used to fill in missing AOD in the MAIAC-Terra AOD maps closest in time to the MISR-RA retrieval by linearly regressing values from a 15 x 15 grid cell region centered on the missing cell value, referred to herein as gap-filled MAIAC. The scatterplots in Fig. S1 show MISR-RA AOD retrievals are higher than those retrieved by MAIAC, and much closer to the AERONET values, for the three case study days with highest AOD. Fig. S1 reinforces the need to scale MAIAC-retrieved AOD before gap-filling MISR-retrieved AOD fields. Based on Fig. S1, a study-specific

AOD filter with an upper bound of 0.4 was used for MAIAC retrievals to reduce potential cloud contamination. Larger gaps caused by cloud contamination in the satellite-retrieved AOD were filled using a 7 x 7 grid cell region of CMAQ-reconstructed AOD value, linearly regressed to the satellite-retrieved AOD. This procedure was repeated multiple times as needed until the SJV study region was filled.

Additionally, we produce gap-filled, aerosol-type-grouped AODs from the original MISR-based AG AODs using the model-based grouped AODs from Step 1, and following the same procedure used for total AOD.



### 3.4 Uncertainty Estimates for Model-Reconstructed and Satellite Total-Column Quantities

Two sets of intermediate analyses are presented where surface-based *in situ* as well as column-integrated observations are provided as ground truth (i.e., their uncertainties are small compared to those of the other values used in this study). First, satellite-retrieved AOD snapshots are evaluated against coincident AERONET observations. Second, a comparison between

daylight-averaged AERONET AOD data, satellite-retrieved AOD snapshots, and model-reconstructed diurnal AOD is presented to determine how well the snapshots represent diurnal values in the study region.

### 3.4.1 Comparison between Satellite-, CMAQ-reconstructed, and Ground-based Total-column AOD Snapshots at Coincident Times

Evaluation of MISR-RA (Limbacher and Kahn, 2014) and MAIAC (Lyapustin et al., 2011b) AOD has been performed

extensively before, but not specifically for the study region, where we have considerable ground-truth data. Overall, there were 14 AERONET sites across the SJV (Fig. 1) during the six case study days. The number of coincident satellite- and ground-AOD observations is dependent on the swath width of each satellite instrument, the retrieval algorithm used, and the polar-orbiting coverage for a given day. Fig. 3 and Table S4 provide scatterplots and a statistical summary, respectively, of AERONET AOD collocated in time and space with the MISR-RA, MAIAC, gap-filled MISR AOD (i.e., $\tau_{FillSAT}$), and

CMAQ results. Although AERONET reports AOD at 550 nm, AOD values at 558 nm were calculated for comparison with the MISR AOD retrievals. Only those Terra MAIAC AOD retrievals that were temporally coincident with MISR-RA retrievals were used in this comparison. A window of ± fifteen minutes was applied to select AERONET measurements as spatiotemporally coincident with the satellite overpass, and corresponding CMAQ hourly, reconstructed AOD values were used.

Overall, the MISR-RA AOD compares well with coincident AERONET AOD, and tends to outperform MAIAC statistically over the SJV across all our case-study days (Table S4). The two best-case days for this analysis are January 20[th] and February 5[th], where AERONET AOD values were relatively high (AOD≥0.15) and there were multiple coincident MISR retrievals across the region. On these days, MAIAC underestimates AOD compared to AERONET, whereas MISR-RA

slightly overestimates AOD. Specifically, for January 20[th] and February 5[th], the MISR-RA-to-AERONET AODs had an overall R of 0.91 and 0.99, and a NME of 0.08 and 0.12, respectively. For MAIAC, the corresponding values are an overall R of 0.66 and 0.93, and a NME of 0.23 and 0.31, respectively.

The comparison of MISR-RA and MAIAC satellite-retrieved AODs with AERONET also illustrates how gap-filling MISR

with scaled and gap-filled MAIAC retrievals produces a more consistent product. For example, the Fig. 3 subplot for February 5[th] shows that gap-filled MISR (i.e., FillSAT) offers better agreement than gap-filled MAIAC at the averaged AERONET retrieved AOD value of 0.47. On this specific day and location there is no coincident MISR retrieval, indicating





that the gap-filled MISR improvement is due to scaled and gap-filled MAIAC used to gap-fill the MISR AOD snapshot. Further evident from Fig. 3, the CMAQ reconstructed values systematically underestimate AOD relative to AERONET in nearly all cases and exhibit greater scatter, hinting at the possible value of applying the measurements as constraints on the model simulations.

5 **3.4.2 Comparison of Satellite-based AOD Snapshots with Daylight-average Ground-based AOD and with Daylight- and Diurnal-average Model-based AOD**

Unlike aerosol radiative forcing, which depends on daytime solar heating, conditions during the full diurnal cycle are relevant for many air quality applications. However, AERONET, as well as the satellites, acquire AOD data only during daylight hours, when the sun is well above the horizon. To test the feasibility of using satellite-based AOD snapshot 10 retrievals as proxies for AOD averaged over daylight hours for the study region, we compare the satellite retrievals (MISR, MAIAC, gap-filled MISR) the with daylight-averaged AERONET-retrieved AOD results (Fig. S2). We subsequently compare the model daylight- and diurnal-average AODs, as well as the AERONET daylight-average AODs, with the respective short-term values from these data sources (Fig. 4) to assess how well snapshot values represent AOD for entire days in the study region. In places where the snapshots are substantially different from the daylight-average or diurnal-15 average AOD values, scaled model results would be required to complete the diurnal air quality picture.

For the initial comparison, all retrieved AERONET values per each of the six case-study days were averaged to obtain a daylight average at each of the 14 sites. For the MISR comparison, we have only the same MISR-RA AOD retrieval snapshots as in Fig. 3. For the study cases, MAIAC can have multiple Terra and Aqua retrievals over the region during one 20 day, occurring at different times, due to the wide MODIS swath. As such, MAIAC Terra-retrieved AOD "coincident" with MISR overpasses are in some cases gap-filled with other scaled-MAIAC Terra/Aqua retrievals acquired during that day. A third satellite-retrieved AOD product is the gap-filled, primarily MISR-derived AOD (*FillSAT*) described in Sect. 3.3. Also shown in Fig. S2 are the CMAQ reconstructed daylight-average AODs, described in Sect. 3.2.

25 Overall, the MISR and FillSAT values are very nearly identical, and they tend to serve as better proxies for the daylight-average AERONET values than CMAQ for the study cases. Table S5 contains a statistical summary of the scatterplot data. For the two best days of January 20[th] and February 5[th], the retrieved AODs for MISR-RA and gap-filled MISR agree better statistically than the other datasets in terms of correlation and error relative to AERONET daylight-average values. Although the retrieved AODs for the MISR-RA and gap-filled MISR slightly outperform MAIAC for the specific case study days, this 30 relationship is likely to change for different domains and time periods. As such, the technique for gap-filling MISR AOD might need to be dynamic in weighting the MAIAC AOD retrievals when applied to other regions. For January 20[th] and February 5[th], the gap-filled MISR-to-daylight-average-AERONET AODs had overall R-values of 0.81 and 0.78, and NME



of 0.16 and 0.28, respectively. This comparison indicates the satellite-retrieved AOD quantities are in sufficient agreement with daylight-averaged ground truth to serve as proxies for the daylight-averaged values during the study period.

A procedure for fusing CMAQ model simulations with surface-based measurements is described briefly in Sect. S3 in the supplemental material, and in detail in Friberg et al. (2016). This procedure was applied to $C_{OBS}$ and $C_{CMAQ}$ (Fig. 2) to produce $C_{FCMAQ}$, also referred to as FCMAQ. The additional step allows us to assess how the spatially extensive satellite data affects the results compared to the model constrained only by local surface observations.

To estimate how well the AOD snapshots might characterize the diurnal-average AOD, diurnal-to-hourly ratios for CMAQ
and FCMAQ are plotted against AERONET retrieved AODs acquired within 15 minutes of the satellite overpasses for each case (Fig. 4 and Table S6). AERONET ratios are plotted as well. The diurnal model and daylight AERONET AOD values are divided by AODs at Terra overpass time within the hour and within 15 minutes for the model and AERONET ratios, respectively. On January 18[th] and 20[th], FCMAQ and daytime CMAQ ratios exhibit the high variability at locations where AERONET ratios were near unity, suggesting that CMAQ diurnal-to-hour ratio are at times spatially biased. But generally,
based on model performance, snapshots acquired at Terra overpass time tend to fall within 10% - 20% of the diurnal-average value, except in some cases when the AOD at overpass time <~0.15 or 0.2. At these smaller AODs, a small absolute change in AOD will produce larger percent changes.

One possible reason for the scatter in Fig. 4 is the model representation of transported aerosol. Transported aerosol above
the boundary layer is dependent on the lower BCs adopted in the model, and thus is not always well represented by CMAQ in this region. For example, the model results indicate minimal vertical distribution of dust aerosol, concentrating all the dust within the planetary boundary layer on the study days, whereas transported dust above the boundary layer is likely to be the major non-spherical aerosol species in this region and season (e.g., Liu et al., 2007b). Any biases in dust AOD retrievals are compounded by inaccuracies in the model-based vertical distributions that are applied during the total-column-to-surface
decomposition step. The impact of errors in the adopted vertical distribution of aerosols on these results, beyond the scope of the current paper, needs to be investigated further. Model aerosol vertical distribution can be further constrained by taking advantage of upwind aerosol elevation retrievals from space-based stereo imaging (MISR), in places where the aerosol sources produce visible plumes, and downwind aerosol layer heights from space-based lidar (e.g., CALIPSO) (Kahn et al., 2008).





### 3.5 Step 4 – Deconstructed Total-column Satellite-measured AOD to Surface PM$_{2.5}$ Mass and Speciated Concentrations

Using CMAQ-based aerosol vertical profiles, near-surface concentrations ($Surface\ C_{FillSAT}^{PM2.5\,FRM}$ and $C_{FillSAT}^{Speciated}$) are obtained from the total-column satellite AOD ($\tau_{FillSAT}$) and aerosol group AOD ($\tau_{FillSAT}^{AG}$) by the following three intermediate steps.

As in previous work, the key step amounts to using model-derived ratios of total-column to near-surface aerosol distributions to obtain near-surface values constrained by total-column measurements (e.g., Liu et al., 2004; Van Donkelaar et al., 2010).

In Eq. (4), the total-column dry particle concentrations for the three aerosol groups ($Columnar\ C_{FillSAT}^{AG}$) are calculated from the AODs, $\tau_{FillSAT}$ and $\tau_{FillSAT}^{AG}$, by reversing the reconstructed extinction process applied to model-only values in Step 2 (Eq.

(3)). The same hygroscopic growth and specific dry scattering or absorbing efficiency factors are used here for consistency. The total-column satellite-based AG concentrations ($Columnar\ C_{FillSAT}^{AG}$) are further stratified into the five total-column representative PM chemical components ($Columnar\ C_{FillSAT}^{Speciated}$), defined in Step 1 according to Eq. (1), using the CMAQ-based species-to-aerosol group partition in Eq. (5). With $Columnar\ C_{FillSAT}^{Speciated}$ defined, satellite-based total-column PM$_{2.5}$ ($Columnar\ C_{FillSAT}^{PM2.5}$) is obtained using Eq. (1). The satellite-derived total-column concentrations are then apportioned to

surface-level concentrations by relying on the vertical distribution of the CMAQ simulations of each species in Eq. (6). The satellite-based surface-level PM$_{2.5}$ concentrations are adjusted to reflect PM$_{2.5\ FRM}$ concentrations using Eq. (2). These relationships were defined in terms of daily AOD and species concentrations.

$$Columnar\ C_{FillSAT}^{AG} = \frac{\tau_{FillSAT}^{AG}}{\int \beta_{dsae,i} f_{RH,i} dz} \tag{4}$$

$$Columnar\ C_{FillSAT}^{Speciated} = Columnar\ C_{FillSAT}^{AG} \left( \frac{Columnar\ C_{CMAQ}^{Speciated}}{Columnar\ C_{CMAQ}^{AG}} \right) \tag{5}$$

$$Surface\ C_{FillSAT}^{Speciated} = Columnar\ C_{FillSAT}^{Speciated} \left( \frac{Surface\ C_{CMAQ}^{Speciated}}{Columnar\ C_{CMAQ}^{Speciated}} \right) \tag{6}$$

### 3.6 Step 5 –Optimized PM$_{2.5}$ FRM and Speciated Concentrations by Fusing Satellite-constrained Values with Ground-monitor Data

The optimized concentration dataset ($C_{OPT}$) closely parallels the surface-measurement-constrained CMAQ simulation described in Eq. (S4). The $C_{OPT}$ dataset is derived by constraining the results with the surface-monitor data near their

locations, and weighting the satellite-constrained concentration values progressively more heavily away from available ground monitors (Fig. 5). Using Eq. (7), the six daily $C_{CMAQ}$ fields in the two-month span are replaced with the satellite-derived daily $C_{FillSAT}$ fields, as these were the days when retrieval conditions were adequate to use the data for the current application (See Sect. 2.1 above). With only 11.5% of the $C_{CMAQ}$ fields changing, the weighting factors ($W$; Eq. (S5)) and





average temporal correlations between the simulations and observations ($R_2$; Eq. (S7)), across all monitors, did not need to be recalculated. Thus, for this study, $C_{OPT}$ diverges from $C_{FCMAQ}$ for 6 days out of the entire 2-month time period.

$$C_{Opt_{s,t}} = \alpha \overline{C_{CMAQ_s}}^{-\beta} \left[ W_{s,t} \left\{ \frac{C_{OBS_{sm,t}}}{\overline{C_{OBS_{sm}}}} \right\}_{krig} + (1 - W_{s,t}) \left\{ \frac{C_{FillSAT_{s,t}}}{\overline{C_{CMAQ_s}}} \right\} \right] \qquad (7)$$

Using the techniques described in the next section, we assess the performance of the optimized surface concentrations in the results section.

### 3.7 Evaluation of Optimized Datasets by Cross-Validation

Three cross-validation techniques are used to evaluate how well the optimized datasets represent diurnal values to identify biases that arise from different sampling frequencies and spatial distribution of monitors across the pollutants. First, a tenfold

10% withholding (10-WH) technique is applied to all species. Then a Leave-One-Out (LOO) cross-validation method is used for all the species with the exception of $PM_{2.5}$. Finally, a Regional Holdout (RH) is used only for $PM_{2.5}$. Brief descriptions of these tests are given here; the results of the tests are discussed in Section 4.2 below.

### 3.7.1 10-fold 10% Holdout Cross-Validation

The dataset performance was evaluated using a tenfold 10% withholding cross-validation analysis. For each of 10

independently run trials, a random 10% of the observations were held back per day and each method ("fused," i.e., surface measurements + model, and "optimized," i.e., surface + satellite measurements + model) was applied to simulate the withheld data. The results from the 10 trials were then combined to provide cross-validation results that allow for the exploration of difference in errors based on proximity to monitors. Across monitors and days, the holdout number corresponds to the number of observations for each pollutant (Table 1), ranging from 44 for $PM_{2.5}$-OC to 779 for $PM_{2.5}$.

### 3.7.2 Leave-one-out Cross-Validation

As an alternative to the 10-WH method, the LOO withholding is applied to the five PM components to overcome the sampling and spatial scarcity. By withholding one location at a time, this location-based cross-validation technique can provide information on how well the CMAQ simulations and satellite-derived concentrations of the fused and optimized datasets, respectively, represent diurnal values at locations further than 50 km from other monitors (see speciated monitor

locations in Fig. 1). With some sites containing more than one monitor, collocated monitors were considered one location, and thus all monitors at a location were withheld for LOO. This cross-validation technique does not provide much insight when the nearest monitor is in close proximity, as is the case with the $PM_{2.5}$ mass monitors.



### 3.7.3 Regional Holdout Cross-Validation

A regional withholding technique is used to evaluate fused and optimized PM$_{2.5}$ datasets, as monitor clustering affect the cross-validation results. For each of the cross-validation regions in Fig. 1, all but one of the monitors in a region is withheld, and this is repeated independently for each daily monitor and region. The approach approximates the evaluation of LOO

when the distance between monitor locations is large (i.e., >50 km).

### 4 Results

Two sets of analyses are presented where surface-based *in situ* observations are provided as ground truth (i.e., their uncertainties are small compared to those of the other values used in this study). First, modeled and deconstructed satellite-constrained results for PM$_{2.5}$ and PM$_{2.5}$ grouped by species are evaluated against EPA AQS and CSN ground observations,

respectively. For the second set of analyses, cross-validation is used to evaluate satellite-constrained model performance. The main objectives of this section are (1) to evaluate the results of Steps 2-5 as much as possible, (for evaluation of Step 1, see Friberg et al. 2017), (2) to assess where, and to what degree, the satellite data help constrain the model PM$_{2.5}$ over an extended region, and (3) where mid-visible AOD values exceed 0.15, to also evaluate the satellite-constrained, speciated PM$_{2.5}$.

### 4.1 Comparison of Satellite-constrained and Model-based Daily PM$_{2.5}$ and Speciated Component Surface Concentrations to Average Daily Ground Truth

We compare now the model-based ($C_{CMAQ}$), model-fused-with-ground-monitor ($C_{FCMAQ}$), deconstructed satellite-constrained ($C_{FillSAT}$) and optimized ($C_{OPT}$; model + ground monitor + satellite) daily averaged PM$_{2.5}$ and speciated component concentrations with EPA AQS and CSN observations. Table S7 provides a statistical summary of the comparison between

the ground truth and the modeled, fused, satellite-constrained, and optimized results, stratified by pollutant, day, and dataset. Fig. 6 presents concentration maps of the four aforementioned datasets with embedded ground truth PM$_{2.5}$ values and their respective RGB images (depicting cloud coverage) for the three days with relatively high AOD in the study set (January 20[th], February 3[rd], and February 5[th]).

Focusing on the area within the SJV, the higher concentration gradients in $C_{FillSAT}$ are due to the application of satellite snapshots. The satellite-constrained concentration snapshots tend to provide more realistic spatial distributions of PM$_{2.5}$ compared to the unconstrained model values, $C_{CMAQ}$. Specifically, the $C_{FillSAT}$ maps show greater dynamic ranges of values, with localized hotspots over known urban areas, such as Bakersfield (35.4° N lat., 119.0° W long.) on January 20[th] and February 5[th], and Fresno (36.7° N lat., 119.8° W long.) on February 3[rd]. The satellite-constrained snapshot results also tend

agree better with available surface measurements in other high-AOD areas, but cloud contamination and the lack of satellite



diurnal sampling affect the $C_{FillSAT}$ values primarily in low-AOD regions. This suggests that the technique will yield increasingly good results when applied in more heavily polluted areas around the globe. Fig. S3 presents scatterplots comparing the daily averaged models and the satellite-constrained snapshots of near-surface PM$_{2.5}$ to ground monitor values. They indicate than diurnal variability is significant in some places and times, but not in others. For high-AOD days (Jan.

20th, Feb. 3rd, and Feb. 5th), Fig. S3 shows $C_{FillSAT}$ PM$_{2.5}$ is in general agreement with surface observations within the performance range of the model results, and the variability is minimal, especially compared to low AOD days. Of the three relatively high-AOD days, January 20th has the least amount of cloud contamination, whereas February 5th has the most. Following the Fig. 5 weighting between the datasets, the visible contributions of the $C_{CMAQ}$ and $C_{FillSAT}$ datasets to the $C_{FCMAQ}$ and $C_{OPT}$ PM$_{2.5}$ fields in Fig. 6 occur at distances of a fifth to a half degree (20 to 50 km) beyond a monitor. At or near a

ground observation, the $C_{OPT}$ fields are weighted towards the interpolated surface-observation fields, whereas the influence of $C_{FillSAT}$ on $C_{OPT}$ improves the regional behaviour and enhances the spatial gradient structure synoptically. For $C_{CMAQ}$ and $C_{FillSAT}$, the estimated temporal variances are fairly constant and do not depend on distance to the surface observations. The surface observations, rather than model or satellite-based results, dominate the $C_{FCMAQ}$ and $C_{OPT}$ temporal correlations at and near monitor locations, whereas $C_{CMAQ}$ and $C_{FillSAT}$ dominate at distances 20 to 50 km beyond a monitor. As such, the

temporal correlations for $C_{CMAQ}$, $C_{FillSAT}$, $C_{FCMAQ}$, and $C_{OPT}$ generally do not approach zero away from the surface stations. For example, on February 5th, the interpolated surface-observation field dominates both the satellite and CMAQ values in the $C_{OPT}$ and $C_{FCMAQ}$ PM$_{2.5}$ maps. The situation at Bakersfield on this day is a bit different. Here the assumed surface monitor uncertainty plays a role, as CMAQ reports a much lower value, the satellite contribution is weighted significantly some distance from the urban center, and the actual difference between the monitor and the $C_{OPT}$ field is about 12.5%, though the

contrast appears large due to the color scale. The satellite contribution is investigated further and quantified in the validation exercises of the next section, where we systematically decrease the dependence of $C_{OPT}$ fields on surface observations.

Figures 7 and S4 provide speciated NO$_3$, NH$_4$, and SO$_4$ surface concentration maps for January 20th and February 3rd; respectively; ground-truth data, available only for February 3rd, are included in Fig. 7. For the evaluation of the modeled and

satellite-constrained surface concentrations, sparse ground observations of speciated PM have a large impact, especially on the high-AOD days. This is compounded by ground-monitor sampling infrequency, as evident in the correlation ranges (Table S7). Fig. S4 demonstrates the ability of satellite aerosol retrievals to characterize the spatial distributions of speciated aerosol airmass types more realistically and consistently than the models across all three species. Unlike for PM$_{2.5}$, there were no speciated monitor measurements available on January 20th, so the OPT results are equal to FillSAT (Fig. S4).

Although the $C_{CMAQ}$ and $C_{FillSAT}$ results show agreement around the locations of known emission sources, the satellite-derived aerosol concentrations at the surface show more realistic horizontal dispersion patterns, and the spatial distribution better reflects the likely influence of topographic features. Specifically, during SJV winters, wide horizontal uniformity of ammonium nitrate concentrations is characteristic of this air basin, due to the near-surface inversion (Watson and Chow,





2002). Particulate nitrate is known to form over non-urban areas when high ammonia emissions from the surface, and nitric acid, formed aloft during night-time decoupling, mix during the morning collapse of the inversion (Watson and Chow, 2002). Throughout the region, consecutive days with low PBL heights are known to produce increased and spatially more uniform concentrations of fine particulate matter, nitrate, and sulfate (Watson and Chow, 2002). The $C_{FillSAT}$ spatial structure

5 and background concentration ranges of 10-15 μg m$^{-3}$ for nitrate and 4-5 μg m$^{-3}$ for ammonium (Fig. S4) reflect the aforementioned concentration dynamics. The differences between the model and satellite-constrained concentration gradients within the SJV are visible on January 20$^{th}$ and February 3$^{rd}$, and the related surface mixing and plume dispersion are evident, especially in Fig. S4. Given the very limited speciated monitor measurements available, the Fig. S5 scatterplots show $C_{FillSAT}$ provides better agreement than the model and fused-model values.

Comparing the results of the current analysis with previous studies that attempt to apply satellite data to surface air quality assessment is a challenge for the following reasons: (1) limited, non-overlapping case study domains; (2) disparity in the spatial resolution at which the analyses are performed, which can bias pixel-to-point comparisons; (3) limited number of ground-truth observations; (4) prevalence of statistics that were averaged over entire seasons or years; (5) lack of actual

surface-concentration statistics reported for the satellite-derived values (i.e., many studies report correlations just between satellite-derived, total-column AOD and surface-based PM$_{2.5}$) and (6) where AOD is the satellite-reported quantity used, algorithm version differences between the AERONET, MISR, and MAIAC products used.

With regard to performance comparisons, the statistical-regression-technique study by Liu et al. (2007b; herein referred to as

Liu2007b) is the most similar to the current analysis. Liu2007b compares 54 ground observations to satellite-derived surface concentrations for PM$_{2.5}$ mass and speciated particles over the western US. The statistical regression technique used 3-hour averaged CTM (GEOS-Chem) results coincident with Terra overpasses for 2005 at 2° by 2.5° spatial resolution. The Liu2007b regression results with removed outliers were as follows: PM$_{2.5}$ R$^2$=0.21, NO$_3$ R$^2$= 0.23, SO$_4$ R$^2$= 0.11, and OC R$^2$= 0.11. In our study, the spatial R$^2$ values for PM$_{2.5}$ averaged 0.53 across all days and 0.73 on Jan. 20$^{th}$, the clearest day

with high AOD. The spatial R$^2$ values for the $C_{FillSAT}$ speciated PM on February 12$^{th}$, the only day for which we have more than one surface measurement, are 0.48 for NO$_3$, 0.10 for SO$_4$, 0.46 for OC, 0.63 for NH$_4$, and 0.41 for EC.

**4.2 Comparison of CMAQ, Fused, and Optimized Datasets to Observed Concentrations**

The model, fused, and optimized datasets are included in the 10-WH cross-validation comparison with the monitor data.

The RMSE, MB, and the spatiotemporal, temporal, and spatial mean correlations for the five datasets are presented in Table S8. The spatiotemporal R$^2$ $C_{OPT\ 10-WH}$ values are 0.79 for PM$_{2.5}$, 0.88 for NO$_3$, 0.78 for SO$_4$, 1.0 for NH$_4$, 0.73 for OC, and 0.31 for EC. The similarities among the PM$_{2.5}$ speciated component 10-WH cross-validation statistics are affected by low





numbers of available observations, sampling frequency, and coincident satellite-retrieval data, particularly for $NH_4$ and EC. As a result, when compared to $C_{CMAQ}$, the $C_{OPT\,10\text{-}WH}$ EC results show a 40% increase in spatial $R^2$ and 10% decrease in spatiotemporal $R^2$, whereas the cross-validation spatiotemporal $R^2$ values for $NH_4$ are biased high. The $SO_4$ spatial and spatiotemporal $R^2$ cross-validation results for both $C_{FCMAQ}$ and $C_{OPT}$ show the largest improvement over the unconstrained

model, with a 43% increase compared to the CMAQ simulation performance. The $PM_{2.5}$ temporal and spatiotemporal $R^2$ cross-validation results are 30% and 13% higher than the CMAQ simulations. The $C_{OPT}$ results from the 10-WH cross-validation would normally provide robust cross-validation results that allow for the exploration of error differences based on proximity to monitors. Overall, the statistical improvement between the CMAQ simulations and cross-validated datasets suggest the empirically based mass reconstruction factors, specific dry efficiencies, and SSA values adopted were adequate

for the SJV domain. The 5-cities study 10-WH cross-validation spatiotemporal $R^2$ ranges were 0.81-0.89 for $SO_4$, 0.67-0.83 for $PM_{2.5}$, 0.52-0.72 for $NO_3$, 0.43-0.80 for $NH_4$, and 0.32-0.51 for OC (Friberg et al., 2017). In light of the 5-cities study, the results for relatively homogeneous pollutants of secondary origin of this study fall within these ranges.

Unlike 10-WH, LOO cross-validation results allow us to leverage the spatial distribution of monitor locations throughout the

domain. Table 2 shows the LOO temporal $R^2$, MB, and RMSE values averaged across monitor locations. The $NH_4$ $C_{OPT\,LOO}$ results improved the most across the $PM_{2.5}$ component species and outperformed temporal $R^2$ for $C_{FCMAQ}$ and $C_{FCMAQ\,LOO}$ values by 10 and 8%. $NH_4$ cross-validation performance is highest for monitor locations closest to the agricultural emission sources in the southern area of domain. This finding agrees with the general expectation that aerosol type uncertainties being lowest when the mid-visible AOD is higher than the threshold of ~0.15. For $SO_4$, the cross-validation for both $C_{FCMAQ\,LOO}$

and $C_{OPT\,LOO}$ datasets show significant improvements in temporal $R^2$ and RMSE. For $NO_3$, temporal $R^2$ of $C_{FCMAQ\,LOO}$ is slightly higher than that of $C_{OPT\,LOO}$, whereas the opposite is true for MB. The OC $C_{OPT\,LOO}$ results are mixed between locations, whereas the EC $C_{OPT\,LOO}$ shows improvements across all locations.

To explore the $PM_{2.5}$ $C_{FillSAT}$ impact of $C_{OPT}$, i.e., combining the surface monitor data to CMAQ simulation plus satellite

results, the spatial cross-validation performance assessment of $PM_{2.5}$ $C_{OPT}$ was expanded to include Regional Holdout (RH), which minimizes the effect of clustered monitors on statistics (Table 3). As expected, removing $PM_{2.5}$ clustered monitors increased the cross-validated datasets reliance of $C_{FCMAQ}$ and $C_{OPT}$ on $C_{CMAQ}$ and $C_{FillSAT}$, thus decreasing temporal $R^2$ values. $PM_{2.5}$ $C_{OPT\,RH}$ results are similar for the $C_{CMAQ}$ and $C_{FCMAQ\,RH}$ datasets, with temporal $R^2$ values of 0.71-0.84 for $C_{FCMAQ\,RH}$ and 0.72-0.83 for $C_{OPT\,RH}$. Improvements in the cross-validation results with respect to CMAQ simulations are observed for

the northern half of the SJV domain, regions 1 and 2 in Fig. 1. Proximity to emission sources, meteorology, and topography contribute to the performance differences between the northern regions 1 and 2, and southern regions 3 and 4. The dominant primary $PM_{2.5}$ mass emission sources (i.e., residential wood combustion, and motor vehicles) as well as the major secondary aerosols in the SJV, are associated with urban hotspots, such as Fresno and Bakersfield (Chen et al., 2007). Winter wind





speeds in the SJV are typically below 4 m s$^{-1}$ (Watson and Chow, 2002). As compared to the southern portion of the SJV, the wind speed is slightly higher and is more consistently southeasterly in the northern part of the domain (Cahill et al., 2011; Hayes et al., 1989). During the winter, regional transport occurs when the nocturnal boundary layer is decoupled from the air aloft; as a result, these higher wind speeds aloft tend not to ventilate the surface, intensifying pollutant surface

concentrations throughout the SJV (Chow et al., 1999), whereas dust originating from desert sources to the east and southeast is likely transported aloft.

In summary these results suggest the optimization method is a viable way of constraining CTM simulations using satellite-retrieved information where ground observations are not available, especially where the AOD is higher than in the SJV cases

available for the current study. Based on these results, including the satellite data improves short- and long-term spatiotemporal air quality metrics for PM$_{2.5}$ mass, and long-term air quality metrics for PM$_{2.5}$ speciated components.

## 5 Conclusions

Even in the best-monitored urban areas, ground-based networks have limited spatial coverage, especially over extended regions downwind of major pollution sources. Building on earlier work that produced a method for fusing surface-based

measurements with model simulations (Friberg et al., 2016; 2017), the current study incorporates satellite-derived AOD and species-related AOD information as additional constraints on the model. The strength of the satellite data is broad spatial coverage, providing radiances that tend to have uniform quality over space and time compared to most suborbital observation datasets. The main limitations are lack of vertical discrimination in most situations, lack of diurnal coverage, and only crude aerosol-type sensitivity, especially at low AOD. The approach presented here uses model simulation along

with surface-based measurements to address these limitations.

Satellite and ground-based aerosol measurements were combined with numerical model simulations to: (1) generate aerosol airmass type maps covering the central California test region for DISCOVER-AQ campaign time period in 2013, (2) explore the viability of using satellite data to improve aerosol airmass type mapping over extended regions, and (3) contribute

regional context to what is known about air pollution sources and trends from point sampling monitors.

Satellites help capture PM$_{2.5}$ distributions over large, under-sampled or un-sampled regions, and its fusion with model results tends to represent spatial gradients better than the unconstrained model. Applied appropriately, satellite data can also improve speciated PM$_{2.5}$ where AOD is sufficiently high (generally mid-visible AOD >~0.15 in the study region). The

satellite-constrained concentration maps are spatially consistent with topography, typifying localized hotspots over known urban areas, and exhibit realistic dispersion patterns in the SJV. Comparison with daylight-averaged AERONET and



diurnally averaged CMAQ modeling demonstrated that, for AOD $>\sim 0.15$ and with outliers removed, the satellite-derived snapshots represent the diurnal values within 10 – 20 % for the study cases. Furthermore, satellite-derived $PM_{2.5}$ was in agreement with surface observations, to within the scatter of unconstrained model results, and variability was reduced on higher AOD days. These results suggest satellite retrievals in conjunction with model results can improve $PM_{2.5}$ spatial

characterization in situations where the AOD is sufficiently high. The satellite aerosol retrievals also represent the spatial distributions of speciated aerosol airmass types more realistically and consistently than the unconstrained model and the model constrained only by surface-monitor data, for nitrate, ammonium, and possibly also sulfate.

For the current study, model-based aerosol vertical distributions were used to address the lack of profile measurements.
However, model aerosol vertical distribution could be constrained on large scales with space-based stereo imaging (e.g., from MISR) near emission sources, at least where plumes are visible in the imagery, and with space-based lidar (e.g., CALIPSO) downwind of sources. Diurnal sampling, the second major limitation in the current satellite application, can be assessed and corrected where needed with a model that has been scaled to available satellite snapshots. Eventually, AOD and possibly speciated AOD from geostationary platforms will provide at least daylight if not fully diurnal values.

Under adequate observing conditions, the technique presented here improves the representation of pollutant spatial distributions in air quality models downwind of emission sources. This physically based method complements statistical approaches, and offers the ability to compare satellite-derived $PM_{2.5}$ and speciated concentrations directly to surface measurements. Although the study domain and timeframe did not offer the high AOD levels where this method would work
best, the SJV offered a substantial quantity of suborbital observations for assessing the results, due to the DISCOVER-AQ campaign.

Expanding this work by applying the technique to the other areas with key ground measurements (i.e., Baltimore DISCOVER-AQ campaign) are possible next steps toward establishing the strengths and limitations of the method. The
technique takes advantage of the stable (i.e., consistent), long-term satellite observations that offer global coverage, and provides speciated constraints based on retrieved microphysical properties for AOD retrievals above about 0.15. Once the aforementioned analyses are completed, the method will likely be applied to a selection of globally distributed urban regions that are downwind of sources, in locations where particulate pollution levels tend to be high.




**Acknowledgements**

We thank the CSRA for creating the WRF meteorological inputs and emissions data used in the various model simulations; the US EPA for creating the CMAQ simulations; the US EPA for establishing and maintaining the AQS and CSN sites used in this work and the AQS datasets; the DISCOVER-AQ project (doi:10.5067/Aircraft/DISCOVER- AQ/Aerosol-TraceGas)
5      for providing some of the field data used in this work; and the NASA AERONET network and its principal investigators, as well as their staffs, for establishing and maintaining the AERONET sites used in this work. The views expressed in this article are those of the authors and do not necessarily represent the views or policies of the U.S. Environmental Protection Agency. The work of M. Friberg is supported under Grant Number NNX13AR89H issued through the NASA Education Minority University Research Education Project (MUREP) as part of the NASA Harriett G. Jenkins Graduate Fellowship
10     program. The work of R. Kahn is supported in part by NASA's Climate and Radiation Research and Analysis Program under H. Maring and NASA's Atmospheric Composition Program under R. Eckman. Last but not least, we thank our anonymous reviewers who provided comments that have helped us refine this paper.



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



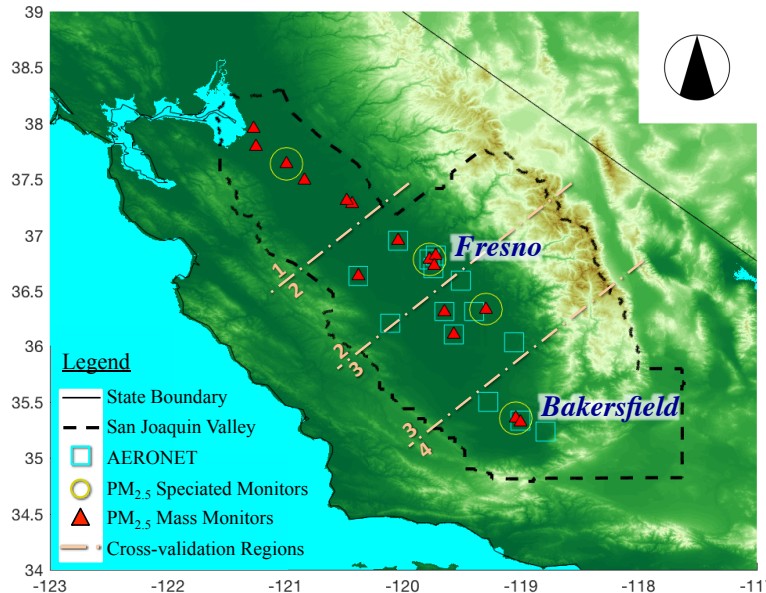

**Figure 1: San Joaquin study area shows the ground elevation, EPA AQS and CSN monitors, and AERONET sites during the two-month NASA DISCOVER-AQ flight campaign.**



**Table 1: EPA AQS and CSN monitor summary statistics for 52 days (6 days).**

| Pollutant | No. of Monitors | Sampling Frequency | OBS | Mean | SD |
|---|---|---|---|---|---|
| $PM_{2.5}$, $\mu g/m^3$ | 22 (21) | 13 daily; 6 1-in-3; 3 1-in-6 | 779 (95) | 21.20 (28.31) | 13.33 (13.51) |
| $PM_{2.5}$-$SO_4$, $\mu g/m^3$ | 7 (6) | 6 1-in-3; 1 1-in-6 | 86 (11) | 0.77 (1.13) | 0.46 (0.69) |
| $PM_{2.5}$-$NO_3$, $\mu g/m^3$ | 7 (6) | 6 1-in-3; 1 1-in-6 | 86 (11) | 7.27 (9.81) | 6.11 (7.38) |
| $PM_{2.5}$-$NH_4$, $\mu g/m^3$ | 5 (4) | 4 1-in-3; 1 1-in-6 | 54 (7) | 2.07 (3.65) | 2.25 (3.32) |
| $PM_{2.5}$-EC, $\mu g/m^3$ | 4 (4) | 3 1-in-3; 1 1-in-6 | 44 (8) | 1.28 (1.14) | 0.77 (0.34) |
| $PM_{2.5}$-OC, $\mu g/m^3$ | 4 (4) | 3 1-in-3; 1 1-in-6 | 44 (8) | 5.25 (5.73) | 3.09 (2.48) |



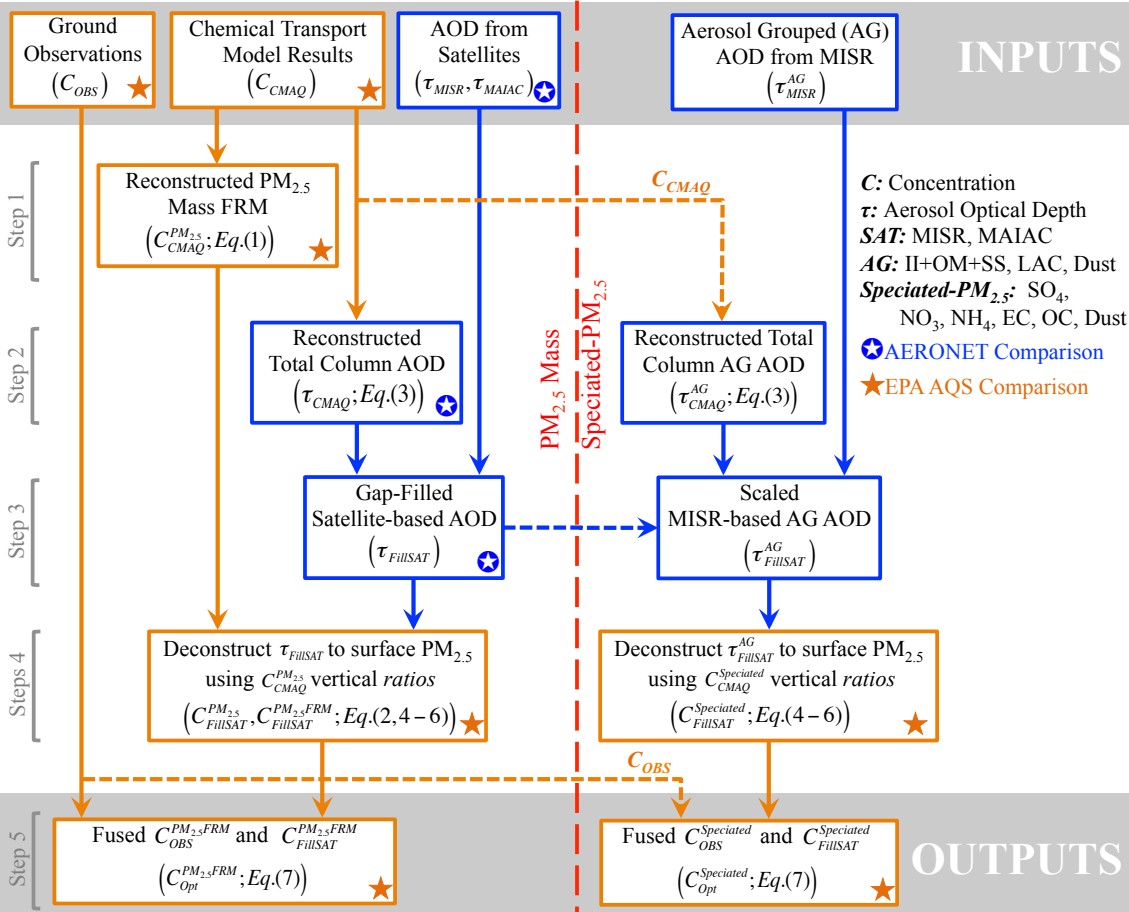

**Figure 2: Methods flow chart connecting satellite retrieved AOD to Modeled AOD, PM2.5 Mass, and PM2.5 Speciated Mass. The parenthetical terms are defined in their respective step in the methods section.**





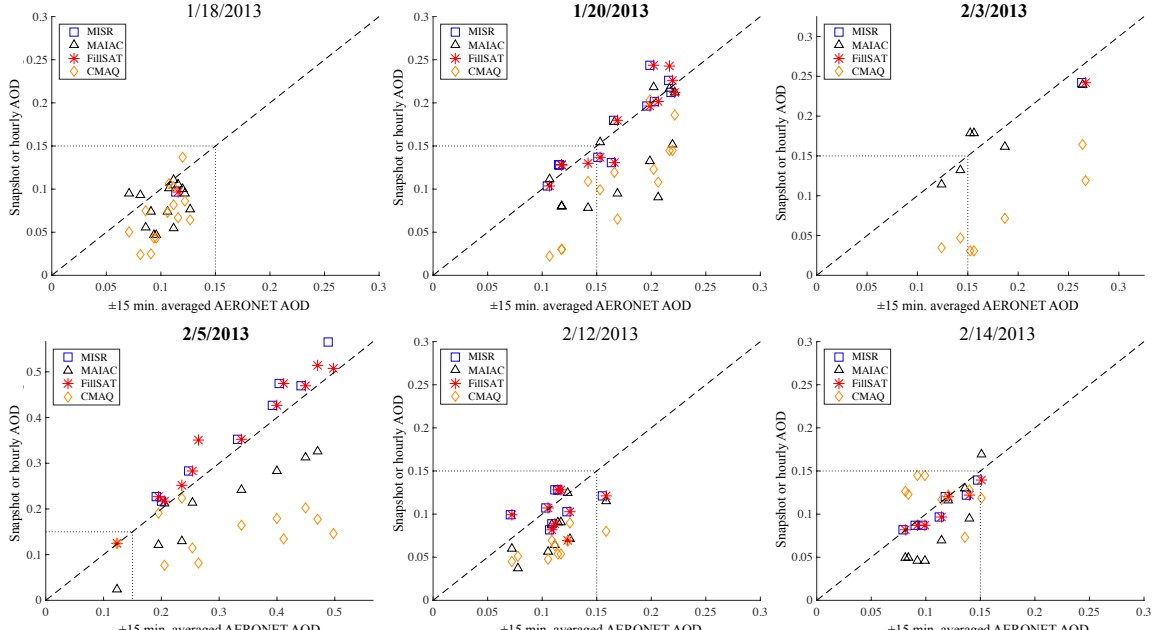

**Figure 3: Scatterplot comparison of AERONET coincidences with MISR, MAIAC, gap-filled MISR, and CMAQ results within ±15 minutes of Terra overpass time. The MAIAC and AERONET AOD comparison is plotted at 550 nm, while the MISR and AERONET AOD comparison is at 558 nm; the dotted line indicates the 0.15 AOD threshold; a 1:1 dashed line is shown for reference.**





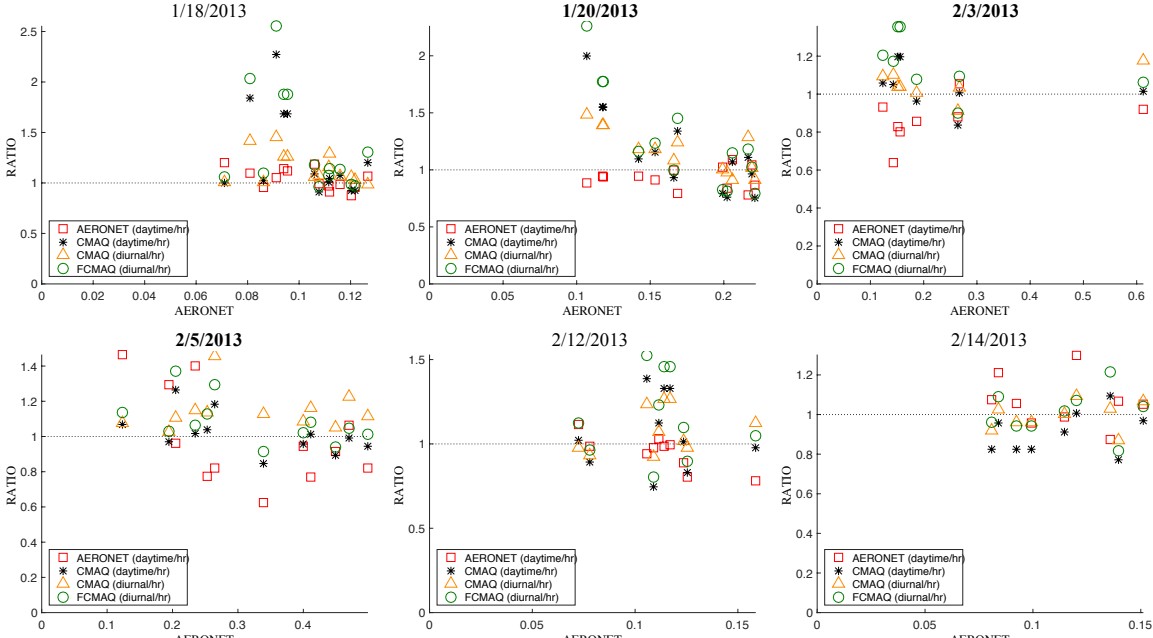

**Figure 4: Scatterplot of daylight-averages to the Terra overpass time ratios versus AERONET AOD retrievals within ±15 minutes of Terra overpass time. Two ratios are shown for CMAQ: daytime average-to-hour ratio and diurnal average-to-hour ratio. The FCMAQ ratios shown are the FCMAQ diurnal to CMAQ hour values. The dashed unity line is included for reference.**





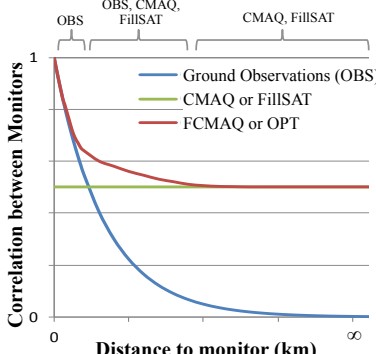

**Figure 5: Theoretical plot of fused and optimized dataset weights as a function of distance from observations. Three regimes identify the contribution of each dataset towards improving concentration field estimates stratified by distance. The exponential decay rate that reflects the temporal Pearson correlation between ground observations as a function of distance is species-specific.**

5   **The temporal variation between ground observations and CMAQ or FillSAT are more consistent (shown as constant), independent of ground observations, and therefore, are not a function of distance to monitor. FCMAQ (surface measurements + model) and OPT (surface + satellite measurements + model) curves show how the strengths of the ground observations and other datasets are maximized using temporal correlations as each grid cell is a function of distance from a ground observation.**



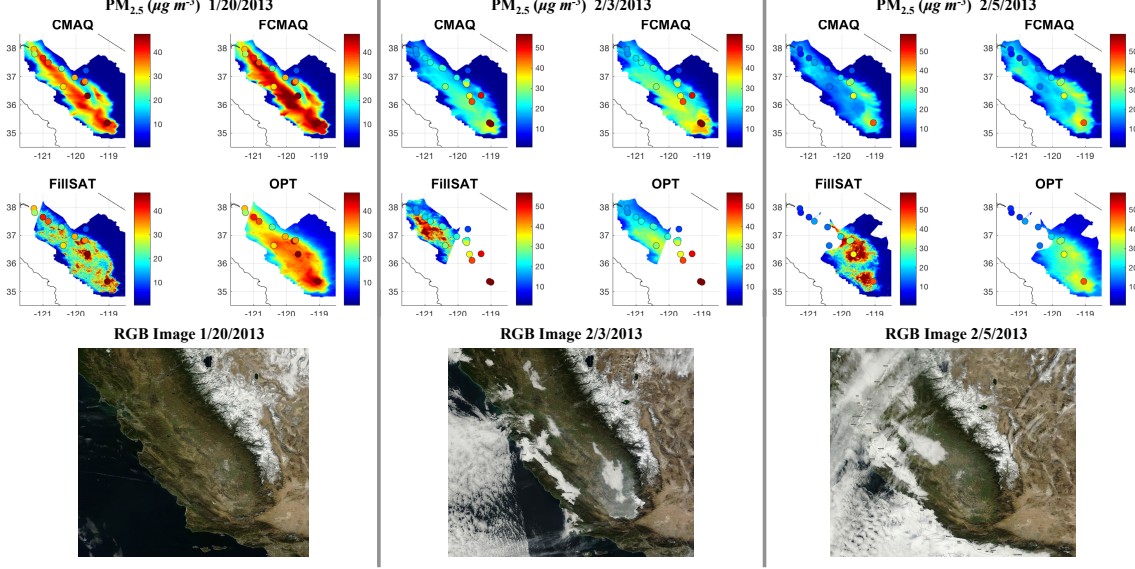

**Figure 6: PM$_{2.5}$ FRM calculated concentration maps with monitor observations (filled circles) and RBG images for the three days with highest AOD. The resolution of the concentration maps are 275 m, whereas the size of the observation markers are ~ 0.1 degrees (~ 11.1 km).**





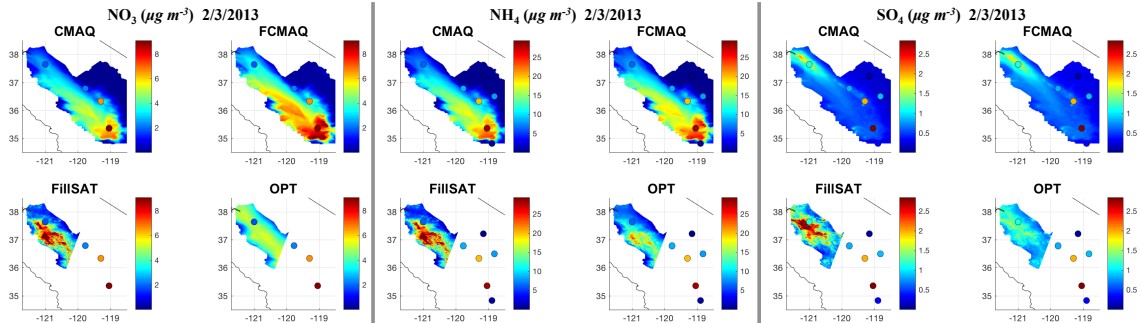

**Figure 7: NH₄, SO₄, and NO₃ calculated concentration maps and monitor observations (filled circles) for February 3ʳᵈ.**





**Table 2: Comparison of averaged temporal R$^2$, mean bias, and root means square error values between observations and leave-one-out cross-validation (LOO CV) for 52 days across all locations.**

| Species | Dataset | Temporal R$^2$ | Mean Bias | RMSE |
|---------|---------|----------------|-----------|------|
| NH4 | CMAQ | 0.52 | 0.43 | 0.94 |
|  | FCMAQ | 1.00 | 0.91 | 1.24 |
|  | OPT | 1.00 | 0.70 | 1.13 |
|  | FCMAQ LOO CV | 0.56 | 0.90 | 1.44 |
|  | OPT LOO CV | 0.62 | 0.71 | 1.39 |
| SO4 | CMAQ | 0.28 | 0.02 | 0.57 |
|  | FCMAQ | 1.00 | 0.00 | 0.12 |
|  | OPT | 0.99 | -0.09 | 0.11 |
|  | FCMAQ LOO CV | 0.75 | -0.06 | 0.41 |
|  | OPT LOO CV | 0.63 | -0.13 | 0.36 |
| NO3 | CMAQ | 0.73 | 0.16 | 0.49 |
|  | FCMAQ | 1.00 | 0.26 | 0.35 |
|  | OPT | 1.00 | 0.12 | 0.31 |
|  | FCMAQ LOO CV | 0.89 | 0.14 | 0.39 |
|  | OPT LOO CV | 0.85 | 0.02 | 0.38 |
| OC | CMAQ | 0.68 | -0.08 | 0.36 |
|  | FCMAQ | 1.00 | -0.11 | 0.14 |
|  | OPT | 1.00 | -0.15 | 0.13 |
|  | FCMAQ LOO CV | 0.68 | -0.12 | 0.34 |
|  | OPT LOO CV | 0.70 | -0.14 | 0.30 |
| EC | CMAQ | 0.52 | 0.31 | 0.53 |
|  | FCMAQ | 1.00 | 0.74 | 0.85 |
|  | OPT | 1.00 | 0.69 | 0.83 |
|  | FCMAQ LOO CV | 0.74 | 0.84 | 0.87 |
|  | OPT LOO CV | 0.76 | 0.80 | 0.88 |



**Table 3: Comparison of temporal R$^2$, mean bias, and root means square error PM$_{2.5}$ values between observations and all simulation, including regional holdout cross-validation (RH CV) for 52 days.**

| PM$_{2.5}$ | Dataset | Temporal R$^2$ | Mean Bias | RMSE |
|---|---|---|---|---|
| Region 1 | CMAQ | 0.68 | 0.17 | 0.40 |
| | FCMAQ | 1.00 | 0.10 | 0.15 |
| | OPT | 1.00 | 0.09 | 0.15 |
| | FCMAQ RH CV | 0.71 | -0.10 | 0.46 |
| | OPT RH CV | 0.73 | -0.12 | 0.46 |
| Region 2 | CMAQ | 0.63 | -0.04 | 0.33 |
| | FCMAQ | 0.99 | 0.05 | 0.18 |
| | OPT | 0.99 | 0.03 | 0.16 |
| | FCMAQ RH CV | 0.75 | 0.05 | 0.33 |
| | OPT RH CV | 0.72 | 0.03 | 0.36 |
| Region 3 | CMAQ | 0.77 | -0.11 | 0.30 |
| | FCMAQ | 1.00 | -0.15 | 0.17 |
| | OPT | 1.00 | -0.17 | 0.17 |
| | FCMAQ RH CV | 0.76 | 0.06 | 0.31 |
| | OPT RH CV | 0.76 | 0.02 | 0.32 |
| Region 4 | CMAQ | 0.82 | -0.11 | 0.34 |
| | FCMAQ | 1.00 | -0.19 | 0.24 |
| | OPT | 1.00 | -0.23 | 0.23 |
| | FCMAQ RH CV | 0.84 | -0.07 | 0.41 |
| | OPT RH CV | 0.83 | -0.11 | 0.39 |