# Peer review of "Constraining Chemical Transport $PM_{2.5}$ Modeling Outputs Using Surface Monitor Measurements and Satellite Retrievals: Application over the San Joaquin Valley"

_Atmospheric Chemistry and Physics, 2018_

## Referee Comment (RC1) · Anonymous Referee #2 · 1 Jun 2018

In this paper, the authors conducted a case study for six days over San Joaquin Valley to constrain model simulated PM2.5 using surface monitor measurements and satellite retrievals. They combined the aerosol products at 275 m spatial resolution from the MISR Research Aerosol retrieval algorithm, ground observations from EPA and the 2 km resolution simulations from WRF/CMAQ to improve the surface estimates of PM2.5, its major chemical component species estimates, and related estimates of uncertainty. The optimized results show good agreements with ground observations for both the total PM2.5 and the species. The method is sound and the results look reliable. I

recommend considering this paper for publication upon response to the following comments:

Major comments:

1. This work is a case study and the authors selected several days with requirements for the MISR data: (1) relatively cloud-free conditions for more MISR coverage; (2) mid-visible AOD exceeds 0.15. They have mentioned in the manuscript that applying this method in other polluted regions are likely to meet common condition with AOD exceeding 0.15. However, what about the coverage issue? For days with limited MISR coverage, the MAIAC AOD used to fill the gap will also have a lot of missing. Then how will this method be applied? This should be discussed in the manuscript.

2. What are the major advantages of this study compared to previous studies that combined information from the satellite retrieval, CTMs and ground observations together? The optimized results in this study seemed not to take advantage of the full coverage of the CTMs.

Minor comments:

1. Page 1, line 30: Why is that EC have much worse performance compared to other species?

2. Page 4, line 1: 1 km or 275 m?

3. Page 6, line 19-20: Will this interpolation process introduce biases?

4. Page 13, line 23: How is the MAIAC AOD scaled before gap-filling MISR AOD? This seems not to be mentioned in the manuscript.

5. In Section 3.4, there are a lot of sentences (e.g. line 25-27 on page 14) reported the evaluation results, which should not belong to the Method section.

6. Figure 6: Although the OPT results had better agreement with ground observations, it still lacks of spatial coverage, even on the selected days with more MISR coverage.

ACPD

---

## Referee Comment (RC2) · Anonymous Referee #1 · 6 Jun 2018

The paper provided a rigorous and detailed analysis of using satellite data (MISR, MODIS), surface observations (AERONET, PM2.5 and aerosol speciation), and CMAQ to derive surface PM2.5 and surface PM speciation. The novelty of this paper, as pointed by the authors, is the use of aerosol type information retrieved from MISR research algorithm. This, however, is really not new, which is also acknowledged in the paper - past work by Liu et al. has used MSIR aerosol type already. The paper also developed several methods for data gap filling, data fusion, and reconstruction of surface PM2.5 and total AOD from CMAQ. To this reviewer, the most interesting part is

indeed the latter, as it has been vague in past studies on how PM2.5 mass is indeed computed with CTM outputs.

The paper has done an excellent job in organizing its structure and presenting the detailed analysis. The paper, however, can be further improved by acknowledging other work done in the past that used satellite observations and CTM together to improve estimate of surface PM2.5. In various places, simplification and summary of the results (from the supplements) can make the paper more easier to read, keep the text flow smoother, improve the clarity, and ultimately enable more readability.

The paper can be published after the following concerns/comments are fully addressed.

General concerns/comments:

1) The title of the paper. The work of this paper in essence is data fusion and statistical analysis by combining data from various sources. While CTM outputs are used, the satellite data here really didn't provide any constraint for improving CTM MODEL-ING that entails emissions, meteorology, different atmospheric processes, and data assimilation. It is recommended to add 'outputs' after 'modeling' in the title to avoid confusion, or change the title to emphasize the data fusion part. This paper didn't improve any components in CTM modeling; instead, it belongs to research of "model output statistics" (MOS) to postprocess model outputs.

2) P2, L3. not sure what 'a systematic and practical approach' means here. As pointed by the first reviewer, there have been much work that combine satellite and groundbased observations already. Please see the summary paper by Hoff and Christopher (2010) prior to 2010 and many other works afterwards. Indeed, the study here is demonstrated for the days and locations that have field campaign data and fewer clouds (compared to many other regions that studied). So, further discussion of the application of the method here in other places is needed. 3) Overall, in what percentage spatially, the AOD values are filled based on MAIA AOD (and scaling factor based on MISR/MODIS AOD ratio)?

Specific concerns/comments:

4) P1, L25. This is a bit confusing. AOD is at 2 km resolution, while aerosol mass type can be retrieved at 275 m resolution? why not AOD at 275 m?

5) P1, L30. R2 is only one of the measures for agreement. How about mean bias and RMSE?

6) P2, L26. Also emissions and parametrization schemes, especially for CTM. See Ge et al., JGR, 2017.

7) P3, L4, it is worth mentioning that early studies, while neglecting these factors (speciation and vertical profile), indeed acknowledge the importance of these factors such as in Wang and Christopher (2003). The current writing gives readers an impression that these early studies didn't recognize the importance of these factors, which in not true. These factors have been recognized since the beginning (Wang and Christopher, 2003).

8) P3, L10. It is worth mentioning that all the work cited here has inconsistence of aerosol optical properties between models and satellite retrieval algorithms. Work has been done that uses CTMs to inform aerosol types for the retrieval from satellites, which in turn improve the estimate of surface PM2.5 from CTM. References include Drury et al. (2010, JGR), Wang et al. (2010, RSE), and van Donkelaar et al., 2013.

9) P5, L5. How long is the DISCOVR-AQ time period? In average, what are the percentage of days that MISR AOD has good spatial converge and AOD is higher than 0.15?

10) P6, L10-20. How many layers in the vertical and in the boundary layer? What is fire emission inventory used? Is CTM outputs data saved at every hour?

11) P7, section 2.5.1. MISR-RA. How does MISR-RA AOD compare with MISR operational AOD? Does MSIR operational product offer the aerosol type retrievals? Using MISR operational product would seem more practical. It will be nice to have some justification here.

12) P11, equation 1. Does CMAQ offer concentration of Al, Si, Ca, Ti, etc? if not, please give the exact equation used in reconstructing CMAQ PM2.5 in this research, so readers don't have to refer to the supplement often.

13) P11, equation 2. How are the values for negative and positive terms in right-hand side of equation obtained in this study?

14) P12, equation 2. fRH (upper case) in equation 3, but frh (lower case) in L15

15) P12, L24. This is not correct. The extinction per unit length is called extinction coefficient, and it is inversely proportional to visibility; see details in Kessner et al., Atmospheric Environment, 2013.

16) P13, and P14; AOD gap filling using MODIS. How to scale MAIA AOD exactly? In cases where both Terra and Aqua MODIS have AOD, is it only Terra MODIS AOD used? Some details are needed here, including when the method works best and when may not work well (such as with large cloud cover).

17) P14, L20-25. it will be good to show a scatter plot that summarizes the comparison for all days in one plot? Also, a plot showing the comparison for data filling only (e.g., in places/times that has no MISR AOD, but filled with MODIS AOD and through scaling/interpolation) can be good to show the improvement by combining both MODIS and MISR.

18) P15, L12. What happens in hours that have cloud? Daily AOD from AERONET has a clear-sky bias.

19) P15, L31 . not sure what 'sufficient' mean here?

20) P15, L11. There are papers talking about diurnal variation of AOD. for example Kaufman et al. in GRL. Are the results here consistent with previous findings?

21) equation 4. This equation is not correct. equation 3 won't give equation 4 as beta, fRH all depends on Z, and C(z) varies with Z.

22) for the results. It will be good to show the summary as several scatter plots respectively for PM2.5 and speciation in all days and sites in the main manuscript. Having summary statistics (such as R, RMSE, and mean bias) in figure. Are the results or improvement by MISR statistically significant?

23) P24, L5. Worthy mentioning recent studies that used VIIRS DNB to derive surface PM2.5 at night. see Wang et al., AE, 2016; Fu et al., 2018.

Again, overall, this is a nice work. it is hoped that the manuscript be further improved by considering the comments above.

---

## Author Comment (AC1) · 13 Jul 2018

**Author Comments (AC) to Referee Comments (RC) 1 – Anonymous Referee 2**

**RC1\_0.** In this paper, the authors conducted a case study for six days over San Joaquin Valley to constrain model simulated  $PM_{2.5}$  using surface monitor measurements and satellite retrievals. They combined the aerosol products at 275 m spatial resolution from the MISR Research Aerosol retrieval algorithm, ground observations from EPA

and the 2 km resolution simulations from WRF/CMAQ to improve the surface estimates of  $PM_{2.5}$ , its major chemical component species estimates, and related estimates of uncertainty. The optimized results show good agreements with ground observations for both the total  $PM_{2.5}$  and the species. The method is sound and the results look reliable. I recommend considering this paper for publication upon response to the following comments:

**AC1\_0.** We thank the reviewer for the valuable comments to improve our manuscript. Please see our itemized responses below.

**Major comments:**

**RC1\_1.** This work is a case study and the authors selected several days with requirements for the MISR data: (1) relatively cloud-free conditions for more MISR coverage; (2) mid-visible AOD exceeds 0.15. They have mentioned in the manuscript that applying this method in other polluted regions are likely to meet common condition with AOD exceeding 0.15. However, what about the coverage issue? For days with limited MISR coverage, the MAIAC AOD used to fill the gap will also have a lot of missing. Then how will this method be applied? This should be discussed in the manuscript.

**AC1\_1.** Where satellite data are missing or where the AOD is too low to provide reliable aerosol type from MISR, we must rely on the emissions-based CMAQ model, tuned, to the extent possible by satellite and surface measurement. Nevertheless, the satellite provides vastly more spatial coverage than the surface stations alone, and this is especially important downwind of major pollution sources. As such, our approach provides improvements where possible, but does not resolve all possible problems. This is now emphasized in the Conclusions section of the paper. The plotting coverage in Figures 6, 7, and S4 has been addressed. Following Figure 5, when FillSAT is not available, the optimized dataset reflects the fused (model + surface measurements) results.
**RC1\_2.** What are the major advantages of this study compared to previous studies that combined information from the satellite retrieval, CTMs and ground observations together? The optimized results in this study seemed not to take advantage of the full coverage of the CTMs.

**AC1\_2.** The physical approach introduced in this study complements the statistical approaches now widely used to take advantage of satellite coverage for air quality applications. Statistical approaches rely on surface-based data training sets to constrain parameters in statistical models, which are then applied elsewhere. Where training data are limited or entirely absent, there is great uncertainty with this approach. In other studies where satellite data are used to constrain a CTM, only the AOD or very limited aerosol-type constraints from the satellite is considered. The physical approach we present makes use of surface data where available, but unlike other approaches relies primarily on both AOD and particle property information contained in the satellite retrievals to constrain a complex, physically based atmospheric dispersion model. This is especially helpful over the vast areas where surface measurements of aerosol concentration and type are not available. We now emphasize this in the Introduction and Conclusions, and mention it in the Abstract.

**Minor comments:**

**RC1\_3.** Page 1, line 30: Why is that EC have much worse performance compared to other species?

**AC1\_3.** Largely emitted from incomplete combustion, EC is a spatially heterogeneous primary species whose particulate phase chemistry and physics is very complex and difficult to model. This is reflected in Table S8, which shows low spatial correlation values and high root mean square error comparison between ground monitors and CMAQ outputs for EC. Appel et al., (2008) discuss overprediction of EC in January and August over western US by CMAQ. EC also relies heavily on the emissions inventory,
and although there have been great strides in the past five years or so to improve EC estimates in the emissions inventory (e.g., residential wood combustion), there are (or at least very likely) still large errors in the inventory relating to EC emissions.

RC1\_4. Page4, line1: 1km or 275m?ĺ AC1\_4. Revised.

**RC1\_5.** Page 6, line 19-20: Will this interpolation process introduce biases? **AC1\_5.** Yes, downscaling CMAQ outputs using any interpolation method inherently introduces biases. Three cross-validation techniques were employed to evaluate the biases of the optimized dataset with respect to ground observations.

**RC1\_6.** Page 13, line 23: How is the MAIAC AOD scaled before gap-filling MISR AOD? This seems not to be mentioned in the manuscript.

**AC1\_6.** We have revised Section 3.3 for clarity as follows:

"To obtain a spatially complete AOD map for each case-study day, we combine the MISR-retrieved, MAIAC-retrieved, and CMAQ-based reconstructed AOD products, as CMAQ can simulate values in all grid boxes, regardless of cloud cover, surface brightness, terrain, and aerosol optical thickness. The most relevant factor affecting spatially complete satellite-retrieved AOD in this study is missing retrievals due to the presence of clouds. The combined AOD product is more complete than the MISR or MAIAC product alone.

The Fig. S1 scatterplots show MISR-RA AOD retrievals are higher than those retrieved by MAIAC, and much closer to the AERONET ground-truth values, for the three case study days with highest AOD. These scatterplots reinforce the need to scale MAIAC-retrieved AOD before gap-filling MISR-retrieved AOD fields. Based on Fig. S1, a study-specific AOD adjustment was applied to the MAIAC data; in addition, a filter with an upper bound of 0.4 was used for MAIAC retrievals to reduce potential cloud

**ACPD**
contamination. On days when Aqua and Terra MAIAC C6v2 AOD retrievals on the 1 km fixed sampling grid were available, the MAIAC-Aqua AOD retrievals were used to fill in missing AOD in the MAIAC-Terra AOD maps (as MAIAC-Terra is closest in time to the MISR-RA retrieval) by linearly regressing values from a 15 x 15 MAIAC-Aqua grid cell region centered on the missing MAIAC-Terra cell value. The 1 km gap-filled MAIAC-Terra AOD maps were subsequently downscaled and spatially interpolated (via bilinear interpolation) to match the downscaled CMAQ 275 m  $\times$  275 m output grid, referred to herein as gap-filled MAIAC. Before combining retrieved AOD products, the 275 m  $\times$  275 m MISR-RA AOD at 558 nm was converted to 550 nm using the retrieved ANG product, and the dynamic sampling grid was re-gridded to match the downscaled CMAQ 275 m  $\times$  275 m grid. The gap-filled MAIAC product was then used to fill in gaps in the MISR-RA AOD product by linearly regressing values from a 15 x 15 gap-filled MAIAC grid cell region centered on the missing MISR-RA cell value. Larger gaps caused by cloud contamination in the satellite-retrieved AOD were filled using a 7 x 7 grid cell region of CMAQ-reconstructed AOD value, linearly regressed to the satellite-retrieved AOD. This procedure was repeated multiple times as needed until the satellite retrieval area within the SJV study region was filled, referred herein as  $\tau_{FillSAT}$ .

A unique component of this work involves the use of the MISR-RA aerosol speciesspecific groups. Consequently, we produce gap-filled, aerosol-type-grouped AODs from the original MISR-based AG AODs using the model-based grouped AODs from Step 1, and following the same gap-filling procedure used for  $\tau_{FillSAT}$ ."

**RC1\_7.** In Section 3.4, there are a lot of sentences (e.g. line 25-27 on page 14) reported the evaluation results, which should not belong to the Method section. **AC1\_7.** The sections 3.4.1 and 3.4.2 comparisons are AERONET validation, critical to the choices made in subsequent steps and, thus, were kept in the Methods section.
**RC1\_8.** Figure 6: Although the OPT results had better agreement with ground observations, it still lacks of spatial coverage, even on the selected days with more MISR coverage.

**AC1\_8.** Please see the response to comment RC1\_1 above.

---

## Author Comment (AC2) · 13 Jul 2018

**Author Comments (AC) to Referee Comments (RC) 2 – Anonymous Referee 1**

**RC2_0**.  The paper provided a rigorous and detailed analysis of using satellite data (MISR, MODIS), surface observations (AERONET, $PM_{2.5}$ and aerosol speciation), and CMAQ to derive surface $PM_{2.5}$ and surface PM speciation.  The novelty of this paper, as pointed by the authors, is the use of aerosol type information retrieved from

MISR research algorithm. This, however, is really not new, which is also acknowledged in the paper - past work by Liu et al. has used MSIR aerosol type already. The paper also developed several methods for data gap filling, data fusion, and reconstruction of surface $PM_{2.5}$ and total AOD from CMAQ. To this reviewer, the most interesting part is indeed the latter, as it has been vague in past studies on how $PM_{2.5}$ mass is indeed computed with CTM outputs.

**AC2_0.** We thank the reviewer for the encouragement and the valuable comments. All the comments have been addressed in the revised manuscript. Please see our itemized responses below.

The approach here is fundamentally different – Liu et al. used a statistical approach, whereas we present here a complementary, physical approach. The underlying model being refined here is the CTM (CMAQ) rather than a regression model. Furthermore, we use as model constraints the particle size and light-absorption information from MISR, in addition to the particle shape, in a novel manner consistent with the limitations of the data. This is now emphasized in the Introduction and Conclusions of the paper.

The paper has done an excellent job in organizing its structure and presenting the detailed analysis. The paper, however, can be further improved by acknowledging other work done in the past that used satellite observations and CTM together to improve estimate of surface $PM_{2.5}$. In various places, simplification and summary of the results (from the supplements) can make the paper more easier to read, keep the text flow smoother, improve the clarity, and ultimately enable more readability. The paper can be published after the following concerns/comments are fully addressed.

**General concerns/comments:**

**RC2_1.** The title of the paper. The work of this paper in essence is data fusion and statistical analysis by combining data from various sources. While CTM outputs are used, the satellite data here really didn't provide any constraint for improving CTM

MODELING that entails emissions, meteorology, different atmospheric processes, and data assimilation. It is recommended to add 'outputs' after 'modeling' in the tittle to avoid confusion, or change the title to emphasize the data fusion part. This paper didn't improve any components in CTM modeling; instead, it belongs to research of "model output statistics" (MOS) to postprocess model outputs.

**AC2_1.** We have added the word "outputs" as recommended.

**RC2_2.** P2, L3. not sure what 'a systematic and practical approach' means here. As pointed by the first reviewer, there have been much work that combine satellite and ground-based observations already. Please see the summary paper by Hoff and Christopher (2010) prior to 2010 and many other works afterwards. Indeed, the study here is demonstrated for the days and locations that have field campaign data and fewer clouds (compared to many other regions that studied). So, further discussion of the application of the method here in other places is needed.

**AC2_2.** Please see the responses AC1_1, AC1_2, and AC2_0. There are fundamental differences with our approach that provide certain advantages. We have made a larger point of the differences and advantages in the revised text.

**RC2_3.** Overall, in what percentage spatially, the AOD values are filled based on MAIA AOD (and scaling factor based on MISR/MODIS AOD ratio)?

**AC2_3.** We gap-fill using results from the MODIS Multi-Angle Implementation of Atmospheric Correction advanced algorithm (MAIAC; Lyapustin et al., 2018). The MISR AOD case study retrievals had 70% or greater spatial coverage of the SJV boundary delineated in Figure 1 and AOD approximately above the 0.15 threshold. Thus, MAIAC (not MAIA) AOD was used to gap-fill between 0 to 30% of the MISR AOD scene for each case. This information is on Page 8 Line 17) in the text.

**Specific concerns/comments:**

**RC2_4.** P1, L25. This is a bit confusing. AOD is at 2 km resolution, while aerosol mass type can be retrieved at 275 m resolution? why not AOD at 275 m?

**AC2_4.** We have removed the phrase "2 km resolution", which referred to the CTM output.

**RC2_5.** P1, L30. R2 is only one of the measures for agreement. How about mean bias and RMSE?

**AC2_5.** We have added RMSE to the abstract. The remaining statistics are reported in Table S8.

**RC2_6.** P2, L26. Also emissions and parametrization schemes, especially for CTM. See Ge et al., JGR, 2017.

**AC2_6.** We revised the sentence as "The accuracy of the simulated fields is also affected by the accuracy of the simulated meteorology, emissions, and of the physical and chemical parameterization schemes specified in the model (Cooke et al., 1999; Tong and Mauzerall, 2006; Monks et al., 2009)" and added citations.

**RC2_7.** P3, L4, it is worth mentioning that early studies, while neglecting these factors (speciation and vertical profile), indeed acknowledge the importance of these factors such as in Wang and Christopher (2003). The current writing gives readers an impression that these early studies didn't recognize the importance of these factors, which in not true. These factors have been recognized since the beginning (Wang and Christopher, 2003).

**AC2_7.** We revised the sentence as "Early space-based $PM_{2.5}$ air quality studies directly correlated satellite-derived AOD from the MODerate resolution Imaging Spectroradiometer (MODIS) instruments and ground-level $PM_{2.5}$ concentrations acknowledged, but did not account for, particle vertical distribution, day-to-day variations,

and/or aerosol speciation (Chu et al., 2003;Wang and Christopher, 2003;Engel-Cox et al., 2004;Chu, 2006;Gupta and Christopher, 2009;Wallace and Kanaroglou, 2007;Schaap et al., 2009;Zhang et al., 2009;Hu and Rao, 2009;Tsai et al., 2011;Hu et al., 2014)" and added citations.

**RC2_8.** P3, L10. It is worth mentioning that all the work cited here has inconsistence of aerosol optical properties between models and satellite retrieval algorithms. Work has been done that uses CTMs to inform aerosol types for the retrieval from satellites, which in turn improve the estimate of surface $PM_{2.5}$ from CTM. References include Drury et al. (2010, JGR), Wang et al. (2010, RSE), and van Donkelaar et al., 2013.
**AC2_8.** We added the sentence "Work has been done to improve estimates of surface $PM_{2.5}$ from CTM by improving the consistency of aerosol optical properties between models and satellite retrieval algorithms, as well as, using CTMs to inform satellite-retrieved aerosol types (Drury et al., 2010; Wang et al., 2010; Donkelaar et al., 2013). However, we map the MISR RA constraints on spherical light-absorbing, spherical non-absorbing, and non-spherical particles to the appropriate aerosol chemical species in the CTM, which is different from previous work." and added citations.

**RC2_9.** P5, L5. How long is the DISCOVR-AQ time period? In average, what are the percentage of days that MISR AOD has good spatial converge and AOD is higher than 0.15?
**AC2_9.** The DISCOVER-AQ SJV deployment ran from 16 January through 08 February 2013. Approximately half of the MISR retrievals met the case requirements of 70% or greater spatial coverage of the SJV boundary delineated in Figure 1 and AOD approximately above the 0.15 threshold. This information has been added to Page 6 Line 3 in the text.

**RC2_10.** P6, L10-20. How many layers in the vertical and in the boundary

layer? What is fire emission inventory used? Is CTM outputs data saved at every hour?

**AC2_10.** The CMAQ domain consisted of 35 vertical layers with varying thickness extending from the surface to 50 hPa and an approximately 10 m midpoint for the lowest (surface) model layer. CMAQ outputs are saved hourly. This information has been added to the text. We used U.S. EPA 2011 NEI emissions data with 2013 updates to fire sources. The wildfire emissions used came from SMARTFIRE v2 (https://www.airfire.org/smartfire/).

**RC2_11.** P7, section 2.5.1. MISR-RA. How does MISR-RA AOD compare with MISR operational AOD? Does MSIR operational product offer the aerosol type retrievals? Using MISR operational product would seem more practical. It will be nice to have some justification here.

**AC2_11.** The current 4.4 km x 4.4 km MISR Standard Algorithm (SA) AOD product was not available at the time of the evaluation and is not available at higher resolution. The SA has greater inconsistencies in aerosol particle retrievals due to limitations in the aerosol climatology included in the algorithm (74 mixtures for the SA vs. over 700 for the RA), poorer surface-reflectance assumptions, issues with the radiometric calibration critical for aerosol-type retrievals that are corrected in the RA, details of the acceptance criteria, and the spatial resolution at which the algorithm is run. For more details, please see the series of papers by Limbacher and Kahn (2014; 2016; 2017; 2018). For particle-type retrievals, the RA performs considerably better than the SA. The information has been added to the text.

**RC2_12.** P11, equation 1. Does CMAQ offer concentration of Al, Si, Ca, Ti, etc? if not, please give the exact equation used in reconstructing CMAQ $PM_{2.5}$ in this research, so readers don't have to refer to the supplement often.

**AC2_12.** Yes, CMAQ does offer these and other dust related concentrations. The aerosol group equations used in this study are included in the supplement for brevity

and are specific to CMAQ v5.0.2.

**RC2_13.** P11, equation 2. How are the values for negative and positive terms in right-hand side of equation obtained in this study?
**AC2_13.** The equation 2 terms are calculated using CMAQ and WRF outputs. These equations are discussed in detail elsewhere (Frank, 2006;Chow et al., 2015) and are referenced appropriately in our paper.

**RC2_14.** P12, equation 2. fRH (upper case) in equation 3, but frh (lower case) in L15
**AC2_14.** We revised the hygroscopic growth factor parameter to $f_{rh}$.

**RC2_15.** P12, L24. This is not correct. The extinction per unit length is called extinction coefficient, and it is inversely proportional to visibility; see details in Kessner et al., Atmospheric Environment, 2013.
**AC2_15.** The sentences were revised as follows:
"The ambient particle extinction as a function of height is the sum of the ambient scattering and absorption with respect to altitude (z), which are the two terms in Eq. (3). When integrated over a horizontal path, the extinction per unit length is sometimes called the visibility, typically reported in Mm-1. From Eq. 3, the dimensionless extinction AOD is obtained by multiplying the ambient particle extinction by the vertical atmospheric path height of each CMAQ layer."

**RC2_16.** P13, and P14; AOD gap filling using MODIS. How to scale MAIA AOD exactly? In cases where both Terra and Aqua MODIS have AOD, is it only Terra MODIS AOD used? Some details are needed here, including when the method works best and when may not work well (such as with large cloud cover).
**AC2_16.** We gap-fill using results from the MODIS Multi-Angle Implementation of Atmospheric Correction advanced algorithm (MAIAC; Lyapustin et al., 2018). Please

see response AC1_6, as Section 3.3 has been revised for clarity.

**RC2_17.** P14, L20-25. it will be good to show a scatter plot that summarizes the comparison for all days in one plot? Also, a plot showing the comparison for data filling only (e.g., in places/times that has no MISR AOD, but filled with MODIS AOD and through scaling/interpolation) can be good to show the improvement by combining both MODIS and MISR.
**AC2_17.** Please see Figures 3, S1, and S2. Aerosol airmass types and spatial distribution change over time. It is not clear to the authors what a scatterplot of all the days in one plot would contribute to the focus of this paper. The individual density scatter plots comparing AERONET, MISR, MAIAC, gap-filled MISR, and CMAQ are shown in Figures S1 and S2.

**RC2_18.** P15, L12. What happens in hours that have cloud? Daily AOD from AERONET has a clear-sky bias.
**AC2_18.** The AERONET clear-sky bias is a limitation of the satellite-based and AERONET AOD comparison. In the optimized dataset the pixels within the domain of study with no satellite-based retrievals rely on the fused CMAQ and ground observations. See also our response to RC1_1.

**RC2_19.** P15, L31 . not sure what 'sufficient' mean here?
**AC2_19.** We have removed the word "sufficient."

**RC2_20.** P15, L11. There are papers talking about diurnal variation of AOD. for example Kaufman et al. in GRL. Are the results here consistent with previous findings?
**AC2_20.** The context of diurnal variation of AOD in the Kaufman et al. (2000) paper is with respect to climate changes and thus compares annual aerosol measurements. The conclusion that a 10:30 AM AOD represents the diurnal average applies to

specific location and/or scenarios. Fire events are some examples of situations where this conclusion does not apply. In our paper, we are interested in capturing diurnal variations with respect to short-term changes (please see Figures 4 and S2). Furthermore, it is difficult to compare diurnal variation conclusions from other study sites due to: (1) the unique weather pattern and pollution transport characteristic of the SJV (i.e., persistent inversion and very low PBL height), (2) differences in product version uncertainty (i.e., AERONET versions between this and earlier studies), and (3) disparity in satellite-retrieved spatial resolution (i.e., biases in earlier studies due to coarser spatial resolution).

**RC2_21.** equation 4. This equation is not correct. equation 3 won't give equation 4 as beta, fRH all depends on Z, and C(z) varies with Z.
**AC2_21.** It is correct that equation 4 depends on altitude. Therefore, we specifically use the height-stratified hygroscopic growth and specific dry extinction efficiency factors from Step 2 for equation 4 in Step 4. Equations 4 through 6 and their descriptions have been updated to make clear these are column-average dry particle concentrations.

**RC2_22.** for the results. It will be good to show the summary as several scatter plots respectively for $PM_{2.5}$ and speciation in all days and sites in the main manuscript. Having summary statistics (such as R, RMSE, and mean bias) in figure. Are the results or improvement by MISR statistically significant?
**AC2_22.** A limitation mentioned in the paper is that the study domain and timeframe did not offer a substantial quantity of suborbital observations for assessing the results statistically. Statistical power is a known issue in this study. Yet, we performed three separate statistical tests to establish as best we could the significance of the results. The comprehensive set of the various summary statistics can be found in the supplemental material.

**RC2_23.** P24, L5. Worthy mentioning recent studies that used VIIRS DNB to derive surface $PM_{2.5}$ at night. see Wang et al., AE, 2016; Fu et al., 2018.

**AC2_23.** The following sentence has been added to the diurnal sampling segment of the conclusion:

"Future research assessing diurnal sampling could benefit from the inclusion of Visible Infrared Imaging Radiometer Suite (VIIRS) instrument datasets, such as daylight-retrieved AOD (Jackson et al., 2013) and Day/Night Band as an estimate of $PM_{2.5}$ surface change (Wang et al., 2016)."

---

## Author Response (AR2)

NASA Goddard Space Flight Center
Code 613, Greenbelt, MD 20771
mariel.d.friberg@nasa.gov

Dr. Qiang Zhang
Co-Editor
*Atmospheric Chemistry and Physics*

July 19, 2018

Dear Dr. Qiang Zhang:

We are pleased to submit our revised manuscript acp-2018-152, titled "Constraining Chemical Transport PM$_{2.5}$ Modeling Output Using Surface Monitor Measurements and Satellite Retrievals: Application over the San Joaquin Valley", for peer-review completion and potential final publication in *Atmospheric Chemistry and Physics* the revised version. We appreciate the referee comment and have updated the manuscript in response the comment.

Thank you for your consideration and assistance.

Sincerely,

Mariel D. Friberg, PhD
Georgia Institute of Technology / NASA Goddard Space Flight Center

Enclosed Files: Cover Letter+Response to Referees+Manuscript with tracked changes, Abstract, Manuscript, and Supporting Information.

**Comment on* "Constraining Chemical Transport PM$_{2.5}$ Modeling Using Surface Monitor Measurements and Satellite Retrievals: Application over the San Joaquin Valley" *by* Mariel D. Friberg et al.**

**Author Comments (AC) to Referee Comments (RC) #2 – Anonymous Referee #1**

**RC2_1.** The authors have addressed most of my comments. One minor revision is needed. "When integrated over a horizontal path, the extinction per unit length is sometimes called the visibility, typically reported in Mm-1". This sentence should be revised. Visibility has unit of length in km or mile. Integration of extinction (Mm-1) with horizontal length (m) will have not unit. Visibility is inversely proportional to extinction. Larger extinction coefficient, lower visibility. Again, feel free to refer to Kessner et al., 2013, Remote sensing of surface visibility from space: A look at the United States East Coast, Atmospheric Environment, 81, 136-147, 2013.

**AC2_1.** The sentence has been removed. Extinction is inversely related to visibility, as the referee says. However, visibility should have the units of length (e.g., you can discern a target having a certain kind of contrast over a distance of x kilometers). Optical depth is unitless, because it reports the ratio of light energy or photons between the beginning and end of a specific path. Extinction cross-section has units of length$^2$, and integrated over a path, you can get visibility in units of length, which is not unitless.

[revised manuscript text omitted]